

# Stratospheric $\delta^{13}CO_2$ observed over Japan and its governing processes and potential as an air age tracer

Satoshi Sugawara[1], Shinji Morimoto[2], Shigeyuki Ishidoya[3], Taku Umezawa[4,2],

Shuji Aoki[2], Takakiyo Nakazawa[2], Sakae Toyoda[5], Kentaro Ishijima[6],

Daisuke Goto[7], and Hideyuki Honda[2]

[1]Miyagi University of Education, Sendai 980-0845, Japan

[2]Center for Atmospheric and Oceanic Studies, Tohoku University, Sendai 980-8578, Japan

[3]National Institute of Advanced Industrial Science and Technology, Tsukuba 305-8569, Japan

[4]National Institute for Environmental Studies, Tsukuba 305-8506, Japan

[5]Institute of Science Tokyo, Yokohama 226-8503, Japan

[6]Meteorological Research Institute, Japan Meteorological Agency, Tsukuba 305-0052, Japan

[7]National Institute of Polar Research, Tachikawa 190-8518, Japan

*Correspondence to*: Satoshi Sugawara (sugawara@staff.miyakyo-u.ac.jp)

**Abstract.** Due to very few reports of $\delta^{13}CO_2$ (the stable carbon isotopic ratio of $CO_2$) observations in the stratosphere, its variations are not well understood. In order to elucidate stratospheric $\delta^{13}CO_2$ variations and their governing mechanisms, and to investigate usefulness of $\delta^{13}CO_2$ as an air age tracer, we have collected stratospheric air samples using balloon-borne cryogenic samplers over Japan since 1985 and analyzed them for $\delta^{13}CO_2$. To obtain precise $\delta^{13}CO_2$ values,

we incorporated the mass-independent fractionation of $^{17}O$ and $^{18}O$ in the $\delta^{13}CO_2$ calculation. $\delta^{13}CO_2$ has decreased through time in the mid-stratosphere with an average rate of change of $-0.026 \pm 0.001$ ‰ $yr^{-1}$ for the period 1985–2020, consistent with that in the troposphere. However, mid-stratospheric $\delta^{13}CO_2$ values did not show a time delay compared to the tropical tropospheric values. This could be explained by the production of $CO_2$ by $CH_4$ oxidation and the gravitational separation of $^{13}CO_2$ and $^{12}CO_2$. To confirm this hypothesis, we used a two-dimensional model to simulate the stratospheric

$\delta^{13}CO_2$ values while accounting for these processes. The results indicate that these two effects strongly impact the vertical distribution of $\delta^{13}CO_2$. We newly defined 'stratospheric potential $\delta^{13}C$' ($\delta^{13}C_P$) as a quasi-conservative parameter incorporating the kinetic isotope effect of $CH_4$ oxidation and gravitational separation, and we used it to estimate the mean age of stratospheric air. Despite large uncertainties, the mean age derived from $\delta^{13}C_P$ was consistent with that derived from the $CO_2$ mole fraction, suggesting its usefulness for further investigation of stratospheric transport processes.

## 1 Introduction

The isotopic ratio of carbon in $CO_2$ (here, we use '$\delta^{13}CO_2$' to distinguish it from that in $CH_4$, '$\delta^{13}CH_4$') is thought to provide information on the carbon cycle, and observations have been carried out mainly using ground stations, ships, and aircraft in the troposphere (e.g. Keeling et al., 1995; Francey et al., 1995; Nakazawa et al., 1993; Morimoto et al., 2000).



However, there have been very few reports of $\delta^{13}CO_2$ observations in the mid-stratosphere, mainly because of the difficulty in collecting air samples at such altitudes; the only available reports are balloon observations over Japan and Scandinavia (Gamo et al., 1995; Aoki et al., 2003). It is well known that tropospheric $\delta^{13}CO_2$ values have decreased with time due to anthropogenic emissions with low $\delta^{13}CO_2$ values resulting from fossil fuel consumption. Gamo et al. (1995) have reported that $\delta^{13}CO_2$ values observed in the mid-stratosphere were higher than those in the troposphere over the same period and that the average stratospheric $\delta^{13}CO_2$ value during 1986–1992 was about −7.5 ‰ and decreasing with time. Aoki et al. (2003) have reported that the mid-stratosphere (25–32 km altitude) $\delta^{13}CO_2$ value over Japan in 1997 was approximately −7.7 ‰; this value is consistent with the expected value extrapolated from the results of Gamo et al. (1995), suggesting that their estimated secular decreasing trend is plausible.

Anomalous $^{18}O$ enrichment in stratospheric $CO_2$ was first reported in balloon observations (Gamo et al., 1989), which spurred extended studies of the mass-independent isotopic effect (MIE) on the triple oxygen isotope system (e.g. Thiemens, 1999; Kawagucci et al., 2008). These MIE studies have clarified that the relationship between $^{17}O$ and $^{18}O$ enrichment in stratospheric $CO_2$ is quite different from that in the troposphere. This fact is very important for stratospheric $\delta^{13}CO_2$ measurements because mass spectrometry measures the ratios $^{45}CO_2/^{44}CO_2$ and $^{46}CO_2/^{44}CO_2$, and an assumed relationship between $^{17}O$ and $^{18}O$ enrichment is needed to calculate $\delta^{13}CO_2$. The influence of the MIE on oxygen in stratospheric $\delta^{13}CO_2$ has not been taken into account in previous studies. Although Aoki et al. (2003) have found that $\delta^{13}CO_2$ values increase with increasing altitude in the stratosphere over Japan and have suggested that the MIE on $^{17}O$ and $^{18}O$ should be considered when interpreting such a vertical distribution, no quantitative investigation of these factors has been conducted.

Stratospheric methane ($CH_4$) is destroyed by reactions with hydroxyl (OH), excited singlet oxygen (O($^1$D)), and chlorine (Cl) radicals, and thus plays an important role in stratospheric chemistry. However, the quantitative contributions of these chemical processes to $CH_4$ loss and their temporal variations remain poorly understood. It is thought that $\delta^{13}CH_4$ provides useful information about $CH_4$ destruction in the stratosphere because characteristic isotopic fractionations occur in each reaction (e.g. Saueressig et al., 1995; Sugawara et al., 1997; Rice et al., 2003; Röckmann et al., 2011). However, only a few measurements have been made so far in the mid-stratosphere, mainly due to the difficulty of collecting air samples at such altitudes. Oxidation of $CH_4$ produces CO, which is eventually oxidized to $CO_2$. In general, the isotopic fractionations in the above reactions cause the faster destruction of $^{12}CH_4$ relative to $^{13}CH_4$, making the chemical products depleted in $^{13}C$ (decreased $\delta^{13}C$) while the remaining $CH_4$ becomes enriched in $^{13}C$ (increased $\delta^{13}CH_4$). Indeed, Brenninkmeijer et al. (1996) have found that CO in the lower Antarctic stratosphere is extremely depleted in $^{13}C$ (decreased $\delta^{13}CO$) mainly because of the local production of CO via the reaction of $CH_4$ with Cl. Recently, Gromov et al. (2018) and Röckmann et al. (2024) have discussed $\delta^{13}CO$ variations associated with the $CH_4$ + Cl reaction. From these studies, it is expected that the $CH_4$–CO–$CO_2$ chain reaction could be an airborne source of $CO_2$ with extremely low $\delta^{13}C$ values, depressing stratospheric $\delta^{13}CO_2$ values. However, no study has yet quantitatively assessed this effect.

The age of stratospheric air is the transit time of air from around the tropical tropopause to a certain location in the stratosphere, which is a powerful tool for diagnosing stratospheric transport (see Garny et al., 2024 for a review of the latest developments in air age studies). It is expected that possible changes to the Brewer–Dobson circulation induced



by climate change are detectable as long-term changes in the mean age of air. For this purpose, age tracers, such as mole fractions of $CO_2$ and $SF_6$, have been measured in air samples collected by scientific balloons and satellites (e.g. Engel et al., 2009, 2017; Stiller et al., 2012). $CO_2$ mole fractions observed in the mid-stratosphere over Japan have been combined with $SF_6$ data to provide the longest record of the mean age of air, as reported by Engel et al. (2009, 2017) and Ray et al.

(2014). Engel et al. (2017) has reported that the mean age of air in the mid-stratosphere does not indicate a significant trend over the period 1975–2016, with only a slightly positive increase of +0.15 ± 0.18 years decade$^{-1}$. However, recent models predict accelerating circulation throughout the stratosphere (e.g. Butchart, 2014), which will decrease the mean age of air. Such a decreasing trend is consistent with observational results in the lower stratosphere (Ray et al., 2014). Despite many efforts to elucidate the trend of age of air, a discrepancy remains between mid-stratosphere age trends in

models and observations. One reason for this discrepancy is the large uncertainties on the observational trends. To resolve this problem, it is necessary to increase the frequency of high-altitude balloon observations and observe additional age tracers beyond $CO_2$ and $SF_6$. Perfluorocarbons and hydrofluorocarbons with extremely long lifetimes could be alternative age tracers (Leedham Elvidge et al., 2018; Laube et al., 2025; Umezawa et al., 2024), and observations of such multiple age tracers should reduce the uncertainties on mean age of air estimates.

The gravitational separation (hereafter, 'GS') of major atmospheric constituents in the stratosphere was first reported from balloon observations (Ishidoya et al., 2006, 2008a, 2008b, 2013). They showed that vertical gradients of the isotopic and elemental ratios of major atmospheric components ($N_2$, $O_2$, and Ar) in the stratosphere are caused by molecular diffusion and depend on molecular mass. Stratospheric GS has now been observed in polar and equatorial regions (Ishidoya et al., 2018; Sugawara et al., 2018) and reproduced in numerical models (Belikov et al., 2019; Birner et

al., 2020). Alongside age of air, stratospheric GS can be used to diagnose stratospheric transport processes (Ishidoya et al., 2013; Sugawara et al., 2018; Birner et al., 2020). These findings also suggest that GS could affect the mole fractions and isotopic ratios of all stratospheric constituents, although no studies have addressed the impact of GS on stratospheric $\delta^{13}CO_2$ values.

The main objective of this study is to clarify the mechanisms governing $\delta^{13}CO_2$ variations in the stratosphere

and understand the long-term variations. A comprehensive understanding of the variations of stratospheric $\delta^{13}CO_2$ requires that the MIE on oxygen, $CH_4$ oxidation and associated isotopic fractionation, mean age of air, and GS be considered. Balloon observations provide a unique opportunity to evaluate these effects by studying a variety of atmospheric constituents using high-precision measurements. In this study, we constructed a new $\delta^{13}CO_2$ record for the stratosphere over Japan spanning the 35 years from 1985 to 2020, primarily by newly analyzing archived $CO_2$ samples and in part by

including previously published data (Gamo et al., 1995; Aoki et al. 2003). We then examined vertical and temporal variations in the record. Furthermore, we investigated in detail the various mechanisms affecting the vertical distribution of stratospheric $\delta^{13}CO_2$ and propose herein a new concept called 'stratospheric potential $\delta^{13}C$'. We also investigated the possibility of estimating the age of stratospheric air from the $\delta^{13}CO_2$ record.

**2 Experimental Procedures**

**2.1 Air sample collection**





We have continuously collected stratospheric air samples over Japan since 1985 using balloon-borne cryogenic samplers (e.g. Nakazawa et al., 1995). A large scientific balloon equipped with a sampler was launched nearly once a year during 1985–2002 from the Sanriku Balloon Center (SBC; 39° 10′ N, 141° 50′ E). The frequency of observations has decreased

since 2004, with the longest interval between observations being 5 years. During this period, the balloon launch site was relocated from SBC to the Taiki Aerospace Research Field in Hokkaido (TARF; 42° 30′ N 143° 26′ E) in 2008. Balloon observations conducted so far over Japan are summarized in Table S1 of the Supplement. The cryogenic air sampler consists mainly of a liquid helium dewar, stainless steel bottles, motor-driven valves, and a control unit (Honda et al., 1996). Liquid helium was used as a refrigerant for collecting stratospheric air cryogenically under low atmospheric

pressures, and 20–30 L STP of stratospheric air was collected in each bottle, depending on the altitude and valve operation. Air sampling was performed typically at 11 different altitudes, spaced about 2 km apart, between 14 and 35 km altitude. Occasionally, anomalously low $CO_2$ mole fractions were observed, probably due to sample deterioration in the bottle. In such cases, both $CO_2$ mole fraction and $\delta^{13}CO_2$ data were excluded from the data record. In 2020, the $CO_2$ mole fraction and $\delta^{13}CO_2$ value at 28.8 and 30.9 km altitude were not available due to water contamination of the air samples.


### 2.2 Analytical methods of mole fractions and isotopic ratios

Collected air samples were distributed to several institutes in Japan for analysis of the mole fractions and isotopic compositions of various gases. The $CO_2$ mole fraction was measured using a nondispersive infrared gas analyzer at Tohoku University (TU) with an analytical precision of <0.02 μmol mol$^{-1}$; the $CO_2$ measurement protocol has been

described in detail in previous studies (Nakazawa et al., 1995; Aoki et al., 2003; Sugawara et al., 2018). $N_2O$ and $CH_4$ mole fractions were measured using gas chromatographs equipped with an electron capture detector and flame ionization detector, respectively, at TU. The $N_2O$ mole fraction was used both for mass spectrometry analyses to determine $\delta^{13}CO_2$ and to investigate the MIE between $^{17}O$ and $^{18}O$ in $CO_2$, as described below. The analytical precisions for the $N_2O$ and $CH_4$ mole fractions were 0.3 nmol mol$^{-1}$ and 3 nmol mol$^{-1}$, respectively (Nakazawa et al., 2002; Aoki et al., 2003; Ishijima

et al., 2007).

The analytical procedures for $\delta^{13}CO_2$ in air samples collected after 1988 have already been described in previous studies (Nakazawa et al., 1993, 1997a; Morimoto et al., 2000, 2006), and is briefly summarized here. An aliquot of each air sample (about 550 mL STP) was introduced into a $CO_2$ extraction system, and the pure extracted $CO_2$ was divided and sealed in small glass tubes; some tubes were archived for future study. The carbon and oxygen isotopic ratios

($\delta^{13}C$ and $\delta^{18}O$) of the sample $CO_2$ were measured using a mass spectrometer (Finnigan MAT-δS) installed at TU against an in-house reference $CO_2$ calibrated to Vienna Peedee Belemnite (VPDB). The analytical precision of $\delta^{13}CO_2$ was ±0.02 ‰ (Morimoto et al., 2000). Air samples collected in 1985, 1986, 1988, and 1989 were analyzed for $\delta^{13}CO_2$ using a mass spectrometer (Finnigan MAT250) with a reproducibility of less than ±0.02 ‰ at the University of Tokyo (Gamo et al., 1989, 1995).

In this study, we modified the calculation of isotopic ratios to take into account the stratospheric MIE and GS. Hereafter, the isotopic ratios of $^{13}C$ to $^{12}C$, $^{17}O$ and $^{18}O$ to $^{16}O$, and $^{45}CO_2$ and $^{46}CO_2$ to $^{44}CO_2$ are expressed as $^{13}R = n(^{13}C)/n(^{12}C)$, $^{17}R = n(^{17}O)/n(^{16}O)$, $^{18}R = n(^{18}O)/n(^{16}O)$, $^{45}R = n(^{45}CO_2)/n(^{44}CO_2)$, and $^{46}R = n(^{46}CO_2)/n(^{44}CO_2)$, respectively,



where $n$ is the amount of each isotopologue. $\delta^{13}CO_2$ is defined as:

$$\delta^{13}CO_2 = \left(\frac{^{13}R}{^{13}R_{VPDB}} - 1\right) \times 10^3 \ (‰), \qquad (1)$$

where $^{13}R_{VPDB}$ is the known $^{13}R$ value of VPDB (Cowplen and Shrestha, 2016). The isotopic ratios $^{45}R$ and $^{46}R$ are

expressed as (Santrock et al., 1985):

$$^{45}R = {}^{13}R + 2 \cdot {}^{17}R \ , \qquad\qquad\qquad (2a)$$

$$^{46}R = 2 \cdot {}^{18}R + 2 \cdot {}^{13}R \cdot {}^{17}R + ({}^{17}R)^2 \ . \quad (2b)$$

To calculate $\delta^{13}C$ and $\delta^{18}O$ values from $^{45}R$ and $^{46}R$, the relationship between $^{17}O$ and $^{18}O$ should be known, because $^{45}CO_2$

contains $^{12}C^{17}O^{16}O$, whereas $^{46}CO_2$ contains both $^{13}C^{17}O^{16}O$ and $^{12}C^{17}O_2$. The relationship between $^{17}R$ and $^{18}R$ can be

expressed as (e.g. Meijer and Li, 1998) :

$$\frac{^{17}R}{^{17}R_{VSMOW}} = \left(\frac{^{18}R}{^{18}R_{VSMOW}}\right)^{\beta}, \qquad\qquad (2c)$$

where $^{17}R_{VSMOW}$ and $^{18}R_{VSMOW}$ are the known ratios of VSMOW (Coplen and Shrestha, 2016). In measurements of

tropospheric $\delta^{13}C$ and $\delta^{18}O$, the mass-dependent relationship between $^{17}O$ and $^{18}O$ can be assumed, i.e. $\beta = 0.528$ (Meijer

and Li, 1998; Assonov and Brenninkmeijer, 2003). However, the MIE between $^{17}O$ and $^{18}O$ is non-negligible in the

stratosphere (Gamo et al., 1995; Thiemens et al., 1999). Therefore, the $\delta^{17}O$ value is essential for accurately determining

the $\delta^{13}C$ value in the stratosphere. Our air samples were also measured for triple oxygen isotopes, as reported by

Kawagucci et al. (2008). However, $\delta^{17}O$ measurements were not performed for all of our stratospheric samples. In such

cases, the following method was applied to obtain $\delta^{17}O$ values. The values of $^{17}R$ and $^{18}R$ calculated from $\delta^{17}O$ and $\delta^{18}O$

observed by Kawagucci et al. (2008) were fitted to Eq. (2c), and the average value of $\beta$ was calculated to be $1.7 \pm 0.2$

using the least squares method. For samples for which $\delta^{17}O$ was not analyzed, we used Eq. (2c) with this $\beta$ value to

calculate $\delta^{13}C$. By substituting Eqs. (2a) and (2c) into (2b), we obtained the following equation:

$$3({}^{17}R)^2 - 2 \cdot {}^{45}R \cdot {}^{17}R - 2\gamma({}^{17}R)^{\frac{1}{\beta}} + {}^{46}R = 0 \ , \qquad (3a)$$

where, $\gamma$ is defined as,

$$\gamma = {}^{18}R_{VSMOW}({}^{17}R_{VSMOW})^{-\frac{1}{\beta}} \ . \qquad\qquad\qquad (3b)$$

Eq. (3a) was solved for $^{17}R$ numerically using the bisection method and the measured $^{45}R$ and $^{46}R$ values, and then $^{13}R$ and

$\delta^{13}C$ were calculated by Eqs. (2a) and (1), respectively. We note that GS occurs in $CO_2$ molecules with different mass

numbers (44, 45, 46). Therefore, it is not accurate to correct $\delta^{13}C$ by assuming a mass number difference of uniquely $\Delta m$

= 1. To obtain the correct $\delta^{13}C$ of stratospheric $CO_2$, it is necessary to first apply the GS correction to the measured values

of $^{45}R$ and $^{46}R$, and then to calculate $\delta^{13}C$ using the method described above. The method used to correct $^{45}R$ and $^{46}R$ for

GS will be discussed in Sect. 3.3.

The MIE calculation described above was not considered in Gamo et al. (1995) and Aoki et al. (2003), and it

is therefore likely that their $\delta^{13}C$ values are not correct, especially at higher altitudes. $\delta^{17}O$ and $\delta^{18}O$ in $CO_2$ increase

rapidly with increasing altitude in the stratosphere (Gamo et al., 1995; Kawagucci et al., 2008), and the deviations from

mass-dependent values also increase with altitude, meaning that the overestimation of $\delta^{13}C$ also increases with altitude.

The influence of the oxygen MIE on $\delta^{13}C$ is almost zero near the tropopause, but at an altitude of about 35 km over Japan,





it is approximately 0.6 ‰. Therefore, this effect significantly impacts the vertical distributions of $\delta^{13}$C. We reanalyzed some of the $CO_2$ samples used by Gamo et al. (1995) and Aoki et al. (2003) to determine the differences between their results (not accounting for the oxygen MIE) and ours to allow us to correct their data and add them to our long-term

record. The MIE on $\delta^{17}$O and $\delta^{18}$O occurs through photochemical reactions in $O_2$, $O_3$, and $CO_2$ (e.g. Thiemens et al., 1999), which are relatively tightly correlated with the mole fraction of $N_2O$, which is also destroyed through photochemical processes. Figure 1 shows the relationships between the $N_2O$ mole fraction and $\delta^{17}$O and $\delta^{18}$O in $CO_2$ in the stratosphere over Japan. These results show that $\delta^{17}$O and $\delta^{18}$O values calculated using $\beta = 1.7$ are consistent with the values observed by Kawagucci et al. (2008).

While cryogenically extracting $CO_2$ from the air sample, $N_2O$ was simultaneously trapped with the $CO_2$. Because $N_2O$ and $CO_2$ have the same mass number, we must correct for $N_2O$ to obtain the true $\delta^{13}$C value. The correction method was described in detail by Nakazawa et al. (1993), and we summarize it here. The ionization efficiency of $N_2O$ was measured in advance, and the measured $\delta^{13}$C value was corrected using the $N_2O/CO_2$ mole fraction ratio of the air sample. In the troposphere, the $N_2O/CO_2$ mole fraction ratio does not change significantly, and the $N_2O$ correction is

almost constant (approximately +0.2 ‰) (Nakazawa et al., 1997a). However, because the $N_2O$ mole fraction decreases rapidly with increasing altitude in the stratosphere (e.g. Toyoda et al., 2001), the $N_2O/CO_2$ mole fraction ratio decreases with altitude, and the magnitude of the $N_2O$ correction also varies vertically. Therefore, the $N_2O$ correction also strongly influences the observed vertical distribution of $\delta^{13}$C in the stratosphere.

        Stratospheric air samples were also measured for $\delta^{13}CH_4$ using a gas chromatograph-combustion-isotope ratio

spectrometer developed at the National Institute of Polar Research (NIPR), Japan. The method has been described in detail by Morimoto et al. (2006) and is briefly summarized here. Each air sample was flushed by pure helium, and the contained $CH_4$ was trapped in a HayeSep D trap at −120 °C. Then, the $CH_4$ was released and transferred into a PoraBOND Q cryofocusing trap at −197 °C. The $CH_4$ released from the trap when brought to room temperature was separated from the residual components using a PoraPLOT Q capillary column and combusted into $CO_2$ in a furnace with a Pt-Ni-Cu

catalyst at 940 °C. The converted $CO_2$ was analyzed using a continuous-flow mass spectrometer (Finnigan MAT252). The precision was reported to be 0.06 ‰ (Morimoto et al., 2006). Before 1996, $\delta^{13}CH_4$ measurements were performed using off-line oxidation equipment (Sugawara et al., 1997), in which $CH_4$ in the sample was converted to $CO_2$ using a catalyst (0.5% Pt supported on alumina pellets) at 750 °C, and the pure $CO_2$ sample thusly prepared was analyzed using the mass spectrometer installed at TU in the same way as for $\delta^{13}CO_2$, except using a micro-volume inlet system. The

overall analytical precision was 0.07 ‰ (Sugawara et al., 1997). To produce a high-quality data record, we needed to verify that these two methods produced the same results. Therefore, several sets of stratospheric air samples were analyzed using both methods, and the $\delta^{13}CH_4$ results differed by about 0.85 ‰. To correct for this difference, the off-line measurement results were adjusted to match the on-line measurements. In this study, $\delta^{13}CH_4$ value is reported relative to VPDB. The international comparison of $\delta^{13}CH_4$ measurements described in detail by Umezawa et al. (2018) revealed a

difference of 0.2 ‰ between NIPR and the Institute of Arctic and Alpine Research (INSTAAR) results. Therefore, all $\delta^{13}CH_4$ values observed by National Oceanic and Atmospheric Administration Earth System Research Laboratories (NOAA/ESRL) and INSTAAR were shifted to match the NIPR scale.





The method used to analyze GS is described in Appendix A.

**3 Results and Discussion**

**3.1 Vertical profiles of $CO_2$ mole fraction and $\delta^{13}CO_2$**

Figure 2 shows vertical profiles of the $CO_2$ mole fraction and $\delta^{13}CO_2$ over Japan throughout the study period. $CO_2$ mole fractions and $\delta^{13}CO_2$ values have increased and decreased, respectively, from 1985 to 2020. Vertical $\delta^{13}CO_2$ profiles observed by Gamo et al. (1995) in 1985 and 1986 are included in this figure after MIE correction. Previous studies reported that the $CO_2$ mole fraction decreases with increasing altitude from the tropopause to around 20–25 km, and then it becomes almost constant at higher altitudes (Nakazawa et al., 1995, 2002). Our $CO_2$ mole fraction data show a similar vertical distribution during the observation period. Because our $\delta^{13}CO_2$ data are more scattered with respect to altitude than the $CO_2$ mole fraction, we calculated detrended vertical $\delta^{13}CO_2$ profiles as follows. First, the long-term trend of the tropospheric $\delta^{13}CO_2$ at Mauna Loa, Hawaii (hereafter, MLO) (White et al., 2024) was calculated as a representative tropospheric trend, and all our stratospheric $\delta^{13}CO_2$ observation data were shifted up or down based on that trend to produce values corresponding to August 2016 for comparison with the model results described below. Then, the $\delta^{13}CO_2$ values thusly obtained for the entire study period were grouped into eleven 2-km vertical bins, and the average value for each bin was calculated (Fig. 2c). The results showed that $\delta^{13}CO_2$ values increased with increasing altitude from near the tropopause to around 25 km and then decreased slightly above that altitude. Gamo et al. (1989) originally reported that $\delta^{13}CO_2$ values increased by approximately 0.4 ‰ from the tropopause to 25 km altitude in September 1985, whereas our recalculated results showed an increase of about 0.2 ‰ (Fig. 2b). The average increase shown in Fig. 2c was even smaller, only about 0.1 ‰. This difference between the vertical profiles originally reported by Gamo et al. (1989) and our present results is due to our incorporation of the MIE on $^{17}O$ and $^{18}O$ (Sect. 2.2).

Below an altitude of ~17 km, $\delta^{13}CO_2$ values are quite variable, often being irregularly high or low depending on the observation year, which is exhibited by the large error bars in Figure 2c. Based on detailed aircraft observations of $\delta^{13}CO_2$ variations in the troposphere over Japan, Nakazawa et al. (1993) have reported that the seasonal cycle of $\delta^{13}CO_2$ between 8 km altitude and the tropopause show a minimum around May and a maximum around September, with a peak-to-peak amplitude of 0.36 ‰. Indeed, our balloon observations were conducted in either spring (late May to early June) or summer (late July to early September), and the observed $\delta^{13}CO_2$ values at lower altitudes tended to be lower in spring and higher in summer. Accordingly, we attribute the large $\delta^{13}CO_2$ variations below an altitude of 17 km to the upper tropospheric air itself, or air directly influenced by it.

Our data clearly reveal a secular decrease in mid-stratospheric $\delta^{13}CO_2$ values. To compare the temporal variations of $\delta^{13}CO_2$ in the mid-stratosphere with those in the troposphere, the $\delta^{13}CO_2$ values obtained by averaging mid-stratospheric data above 24 km altitude ($\lesssim 30$ hPa) from each year and the continuous MLO data are shown in Figure 3. When comparing our data with the NOAA/ESRL and INSTAAR data at MLO (White et al., 2024), any possible offset of $\delta^{13}CO_2$ measurements should be taken into account. The inter-comparison results of the WMO/IAEA Round Robin Comparison Experiment (https://gml.noaa.gov/ccgg/wmorr/; accessed 20 Feburary 2025) show that our values are approximately 0.07 ‰ lower than INSTAAR values. In this study, we directly compared $\delta^{13}CO_2$ data at Ny-Ålesund,





Svalbard observed by Goto et al. (2017) with the INSTAAR data at the Zeppelin Observatory (ZEP) in Ny-Ålesund

(White et al., 2024) because we used the same $\delta^{13}CO_2$ measurement system and standard scale as Goto et al. (2017).

Monthly average $\delta^{13}CO_2$ values at Ny-Ålesund from both studies were calculated by applying a curve-fitting procedure

(Nakazawa et al., 1997b) over the years 1996–2014 compared. As a result, we estimated the difference between this study

and INSTAAR data to be insignificant ($-0.005 \pm 0.067$ ‰) and did not apply any further correction to our $\delta^{13}CO_2$ data.

Annual average $\delta^{13}CO_2$ values at MLO were calculated by applying the same curve-fitting procedure as described above.

The average rate of change of $\delta^{13}CO_2$ in the mid-stratosphere was calculated to be $-0.026 \pm 0.001$ ‰ $yr^{-1}$ for the period

1985–2020, consistent with the average rate of change at MLO from 1990 to 2022 ($-0.027 \pm 0.000$ ‰ $yr^{-1}$) despite the

slightly different time periods. Tropospheric $\delta^{13}CO_2$ values are known to decrease over time due to the emission of $CO_2$

with low $\delta^{13}CO_2$ by anthropogenic fossil fuel consumption, and our results show that a similar decreasing trend is

observable in the stratosphere. It is also well known that mid-stratospheric air over the mid-latitudes is 'aged', mainly

because of the slow transport of air from the tropical tropopause to the mid-latitude mid-stratosphere. Indeed, the age of

mid-stratospheric air estimated from $CO_2$ mole fractions was approximately 4.4 years older on average than that of

tropical tropospheric air (see Sect. 3.5). Nevertheless, the $\delta^{13}CO_2$ values observed in the mid-stratosphere over Japan did

not seem to show any significant time delay compared with the observed values in the tropical troposphere. Although

systematic observations of $\delta^{13}CO_2$ have not yet been performed in the tropical upper troposphere where stratospheric air

originates, Assonov et al. (2010) have found from CARIBIC aircraft observations that $\delta^{13}CO_2$ variations in the tropical

upper troposphere are close to those at MLO. Therefore, if we assume no difference between the $\delta^{13}CO_2$ values at MLO

and those in the tropical upper troposphere, the lack of a delay between the values in the mid-stratosphere over Japan and

those at MLO clearly contradicts the concept of the age of air. Simply following the logic of the age of air, we would

expect the decrease of stratospheric $\delta^{13}CO_2$ to lag behind the tropospheric trend, such that the $\delta^{13}CO_2$ value at any given

time should be higher in the mid-stratosphere than in the troposphere. This discrepancy suggests that $\delta^{13}CO_2$ is not a

conserved quantity in the stratosphere and is instead modified not only by air transport but also by other mechanisms. In

the following sections, we discuss possible mechanisms in detail.

**3.2 Airborne $CO_2$ source from $CH_4$ oxidation**

In the stratosphere, $CH_4$ is destroyed by reactions with OH, $O(^1D)$, and Cl, which produce intermediate products such as

$CH_3O_2$ and $CH_2O$. These products are immediately converted to CO because of their short lifetimes. However, CO is also

destroyed by reaction with OH. Therefore, the chemical budget of CO in the stratosphere is expressed as (Minschwaner

et al., 2010):

$$\frac{d[CO]}{dt} = [CH_4]\big(k_{CH_4+OH}[OH] + k_{CH_4+Cl}[Cl] + k_{CH_4+O}[O(^1D)]\big) + J_{CO_2}[CO_2] - k_{CO+OH}[CO][OH] , \quad (4)$$

where $k_{CH_4+OH}$, $k_{CH_4+Cl}$, $k_{CH_4+O}$, and $k_{CO+OH}$ are the reaction rates for the respective chemical reactions, and $J_{CO_2}$ is the

rate of photodissociation of $CO_2$. The last term on the right-hand side of this equation represents the oxidation of CO to

$CO_2$. Although the amount of $CO_2$ produced this way is small compared to the stratospheric abundance, it cannot be

ignored as an airborne source, especially in terms of isotopic signatures. The timescales for the photochemical loss and

production of CO in the mid-latitude mid-stratosphere (~35 km altitude) are about 30 days (Minschwaner et al., 2010),



quite short compared to the timescale of the stratospheric mean meridional circulation (more than several years). If we assume that the chemical budget of CO is in a steady state and that $CO_2$ photodissociation is negligibly small below the mid-stratosphere, the rate of $CH_4$ loss should equal the rate of $CO_2$ production. Indeed, to estimate the age of stratospheric air from the $CO_2$ mole fraction, a small amount of $CO_2$ produced by the chemical destruction of $CH_4$ was taken into account (e.g., Engel et al., 2002; Sugawara et al., 2018). To correct for this effect, each observed $CO_2$ mole fraction was

adjusted using the $CH_4$ mole fraction measured from the corresponding air sample prior to age calculation (see Appendix C). The adjustment increases with increasing altitude and becomes larger than 1 µmol mol$^{-1}$ at the highest altitudes. As with the $CO_2$ mole fraction, $^{13}C$ can be corrected by assuming a steady state $^{13}CO$ budget in which $^{13}C$ is added to $^{13}CO_2$ from $^{13}CH_4$ through a series of oxidation reactions from $CH_4$ to $CO_2$. Indeed, it is well known that isotopic fractionation occurs during $CH_4$ oxidation (Saueressig et al., 1995; Sugawara et al., 1997; Rice et al., 2003; Röckmann et al., 2011).

Figures 4a, b shows the vertical $CH_4$ mole fraction and $\delta^{13}CH_4$ profiles observed over Japan, respectively. A common feature of both profiles is their rapid change with altitude, although $\delta^{13}CH_4$ values increase, whereas the mole fractions decrease with increasing altitude. Such vertical distributions have been attributed to large kinetic isotopic effects (KIEs) associated with the chemical destruction of $CH_4$ in the stratosphere (Sugawara et al., 1997; Rice et al., 2003; Röckmann et al., 2011). KIEs on $CH_4$ have also been studied using numerical models (Saueressig et al., 2001; Wang et

al., 2002; McCarthy et al., 2003). In previous studies, the apparent fractionation factor was derived from the Rayleigh distillation model by assuming that the chemical destruction of $CH_4$ occurs in a closed system (Sugawara et al., 1997; Rice et al., 2003). In general, the Rayleigh distillation model for the $CH_4$ mole fraction and $\delta^{13}CH_4$ is expressed as:

$$\delta^{13}CH_4 = \left(\delta^{13}CH_{4\_0} + 1\right)\mu^{\alpha-1} - 1 , \quad (5)$$

where $\delta^{13}CH_{4\_0}$, $\mu$, and $\alpha$ are the isotopic ratio before $CH_4$ is destroyed, the ratio of mole fractions ($\mu$ =[$CH_4$]/[$CH_4$]$_0$),

and the apparent fractionation factor, respectively. The apparent fractionation factor can be estimated by applying Eq. (5) to observed data. We note that [$CH_4$]$_0$ and $\delta^{13}CH_{4\_0}$ change with time and should be taken here as the values in the tropical upper troposphere at the time when air entered into the stratosphere through the tropical tropopause layer (TTL). Therefore, the tropical upper tropospheric records of $CH_4$ and $\delta^{13}CH_4$ were used to give [$CH_4$]$_0$ and $\delta^{13}CH_{4\_0}$ values going back in time based on the mean age estimated from $CO_2$ mole fractions (see Appendix C). Details of the tropical upper

tropospheric records of $CH_4$ and $\delta^{13}CH_4$ are described in Sect. 3.5. Figure 4c shows that the relationship between $\ln\{(\delta^{13}CH_4 + 1)/(\delta^{13}CH_{4\_0} + 1)\}$ and $\ln([CH_4]/[CH_4]_0)$ can be approximated by linear functions if we divided the data between the lower (<24 km ) and mid-stratosphere (>24 km ), with the upper layer showing larger isotopic fractionation effects. The apparent fractionation factors calculated using the least-squares method were 0.9889 ± 0.0003 and 0.9866 ± 0.0007 for the lower and mid-stratosphere, respectively. We attributed these differences in apparent fractionation factors

to differences in the relative contributions of the $CH_4$ reactions with OH, O($^1$D), or Cl over the air transport pathway, and/or the effect of air mixing (Röckmann et al., 2011).

Eq. (5) expresses the relationship between the mole fraction and isotopic ratio of the remaining $CH_4$. Similarly, focusing on the $CH_4$ lost via the $CH_4$–CO–$CO_2$ chain reaction, the relationship between the $CH_4$ mole fraction and isotopic ratio of lost $CH_4$ (hereafter, $\delta^{13}CH_{4\_L}$ is derived from Eq. (5) as:

$$\delta^{13}CH_{4\_L} = \left(\delta^{13}CH_{4\_0} + 1\right)\frac{1-\mu^{\alpha}}{1-\mu} - 1. \quad (6)$$



Here, the removed $CH_4$ includes the total amount of $CH_4$ loss integrated over the air transport pathway. Based on this equation, we calculated $\delta^{13}CH_{4\_L}$ for each year's data from the observed $CH_4$ mole fractions and apparent fractionation factors by grouping them into altitudes below and above 24 km. Typical $\delta^{13}CH_{4\_L}$ values were about −57, −56, and −55 ‰ around altitudes of 20, 30, and 35 km, respectively. The stratospheric $\delta^{13}CO_2$ values were corrected for airborne sources

simply by assuming that carbon with the isotopic ratio of $\delta^{13}CH_{4\_L}$ was added to ambient $CO_2$. We note that the amount of $CO_2$ added by $CH_4$ oxidation was estimated from the difference between the $CH_4$ mole fraction observed in the mid-latitude stratosphere and that estimated at the tropical upper troposphere; this does not represent chemical production at the observed location, but apparent production, because $CH_4$ oxidation occurs along the transport pathway of the air mass and is accompanied by air mixing. Consequently, the depression of $CH_4$ integrated over the air transport pathway is

observed at a certain location in the stratosphere. Because we assumed simple Rayleigh distillation model, the $\delta^{13}CH_{4\_L}$ estimated here is also apparent value in a similar sense as described above. Accordingly, our assumption described above was a first-order approximation to correct $\delta^{13}CO_2$ for airborne $CO_2$ sources.

As an example, we corrected the vertical distributions of the $CO_2$ mole fraction and $\delta^{13}CO_2$ (Fig. 5a) and compared with the $CH_4$ mole fraction and $\delta^{13}CH_4$ (Fig. 5b) observed on 22 August 2010. The results indicated that the

$\delta^{13}CO_2$ value decreased by 0.14 ‰ and the $CO_2$ mole fraction increased by approximately 1.1 μmol mol$^{-1}$ at the highest altitude (~34 km) when corrected for airborne $CO_2$ sources. Therefore, such a correction for the airborne $CO_2$ source with low $\delta^{13}CO_2$ values produced by $CH_4$ oxidation is essential for understanding $CO_2$ mole fraction and $\delta^{13}CO_2$ variations in the mid-stratosphere. This effect is particularly strong on $\delta^{13}CO_2$ and significantly impacted the shape of the vertical distribution.

We note that the transport of CO from the troposphere into the stratosphere and its oxidation in the lower stratosphere are not considered in the above discussion. Because the photochemical timescales of CO are longer in the lower stratosphere than in the mid-stratosphere, tropospheric CO could be a stratospheric $CO_2$ source associated with troposphere–stratosphere exchange and may be important in the lower stratospheric CO budget. Observational results on the stable carbon isotopic ratio of CO, $\delta^{13}CO$, have been reported not only for the troposphere (Brenninkmeijer, 1993;

Röckmann and Brenninkmeijer, 1997; Röckmann et al., 1999, 2002; Kato et al., 2000) but also for the lower stratosphere (Brenninkmeijer et al., 1996; Röckmann et al., 2024). In particular, Brenninkmeijer et al. (1996) have found that extremely low $\delta^{13}CO$ values in the southern high-latitude lower stratosphere are accompanied by increasing $\delta^{13}CH_4$ values and have suggested that the large KIE of $CH_4$ destruction by Cl plays an important role. The local production of CO by $CH_4$ destruction depleted $^{13}C$ in stratospheric CO is consistent with the above discussion if CO is rapidly oxidized to $CO_2$

depleted in $^{13}C$. However, tropospheric CO could be an additional airborne $CO_2$ source if it is transported into the lower stratosphere and oxidized to $CO_2$. Previous studies have reported that $\delta^{13}CO$ values in the free troposphere are around −30 ‰ to −25 ‰ (Brenninkmeijer, 1993; Röckmann and Brenninkmeijer, 1997; Röckmann et al., 1999, 2002; Kato et al., 2000), and the KIE on carbon in the reaction CO + OH depends on atmospheric pressure (Stevens and Wagner, 1989; Bergamaschi et al., 2000). Its fractionation factor is 0.994 (expressed as the ratio of rate constants, $^{13}k/^{12}k$) at 1013 hPa,

but increases to about 1.000 at 400 hPa and further to 1.003–1.005 above the tropopause. Therefore, $CO_2$ produced from the reaction CO + OH should be slightly enriched in $^{13}C$ in the stratosphere, contrary to $CO_2$ produced in the lower





troposphere. Accordingly, tropospheric CO could be an airborne $CO_2$ source with $\delta^{13}C$ values lower than around $-25$ ‰ to $-20$ ‰ in the lower stratosphere. Assuming that tropospheric air containing 250 nmol mol$^{-1}$ CO with $\delta^{13}CO = -30$ ‰ is transported to the stratosphere where 200 nmol mol$^{-1}$ CO is lost by reaction with OH, $\delta^{13}CO_2$ values should decrease

by only about 0.01 ‰ when 0.2 µmol mol$^{-1}$ $CO_2$ with $\delta^{13}CO_2 = -28$ ‰ is added to 390 µmol mol$^{-1}$ $CO_2$ with $\delta^{13}CO_2 = -8.5$ ‰. Therefore, this effect would be negligibly small compared with the influence of $CH_4$ oxidation. Unfortunately, we could not further evaluate the effect of tropospheric CO from our observations, and further study including measurements of $\delta^{13}CO$ is needed.

**3.3 Gravitational separation of $\delta^{13}CO_2$**

The standardized GS is defined as $<\delta_G>$, which is normalized by the mass number differences of corresponding molecules (see Appendix A). With respect to $\delta^{13}CO_2$, the upward decrease due to GS is equal to $<\delta_G>$ for $^{13}C^{16}O_2$ and $^{12}C^{17}O^{16}O$ ($\Delta m = 1$), and is 2 times larger for $^{12}C^{18}O^{16}O$, $^{13}C^{17}O^{16}O$, and $^{12}C^{17}O_2$ ($\Delta m = 2$). Therefore, GS equal to and twice the magnitude of $< \delta_G>$ can be considered to act on $\delta^{45}CO_2$ and $\delta^{46}CO_2$, respectively:

$$\Delta_G(\delta^{45}CO_2) = \langle\delta_G\rangle - \langle\delta_G\rangle_0 \, , \qquad (7a)$$

$$\Delta_G(\delta^{46}CO_2) = 2(\langle\delta_G\rangle - \langle\delta_G\rangle_0) \, . \qquad (7b)$$

Here, $\Delta_G(x)$ represents the change in isotopic value $x$ due to GS. Because $<\delta_G> - <\delta_G>_0$ is negative in the stratosphere (see Appendix A), $\Delta_G(\delta^{45}CO_2)$ and $\Delta_G(\delta^{46}CO_2)$ are also negative. When $\delta^{13}CO_2$ was corrected for GS, Eqs. (7a) and (7b) were first applied to $\delta^{45}CO_2$ and $\delta^{46}CO_2$, respectively, and then $\delta^{13}CO_2$ was determined using the MIE algorithm described

in Sect. 2.2. As shown in Figure 5c, $|\Delta_G(\delta^{13}CO_2)|$ was slightly smaller than (about 82% of) $|\Delta_G(\delta^{45}CO_2)|$ in the mid-stratosphere, mainly because of $^{12}C^{17}O^{16}O$ enrichment due to the MIE in the stratosphere.

The same method can be applied to the mole fraction of a specific molecule as (Ishidoya et al., 2006):

$$\Delta_G(C) = C_0 \times (m - m_{air}) \times (\langle\delta_G\rangle - \langle\delta_G\rangle_0) \quad . \qquad (8)$$

Here, $C_0$ and $C$ denote the mole fractions of the molecule before and after GS, respectively, and $m$ and $m_{air}$ are the

respective mass numbers of the molecule and air. The maximum depression of the $CO_2$ mole fraction amounts to about 0.6 µmol mol$^{-1}$ at the highest altitude, assuming $C_0 = 400$ µmol mol$^{-1}$ and $<\delta_G> - < \delta_G >_0 = -100$ per meg. As an example, the vertical profile of the $CO_2$ mole fraction obtained by correcting the 22 August 2010 observations for GS is shown in Figure 5a. Compared to the profiles corrected for airborne $CO_2$ sources, GS correction increases the $CO_2$ mole fraction by 0.4 µmol mol$^{-1}$ and the $\delta^{13}CO_2$ by approximately 0.06 ‰ at 34 km altitude. Although the effect of GS on the $CO_2$ mole

fraction is small, that on $\delta^{13}CO_2$ is non-negligible, significantly influencing the vertical profile along with $CH_4$ oxidation.

Correcting for both $CH_4$ oxidation and GS, $\delta^{13}CO_2$ values increase with altitude and become nearly constant above ~25 km; they are clearly anticorrelated with the $CO_2$ mole fraction. Assuming a linear relationship between the $CO_2$ mole fraction and $\delta^{13}CO_2$ (Fig. 5d), the slope calculated by least-squares is $-0.005 \pm 0.003$ ‰ (µmol mol$^{-1}$)$^{-1}$ for the raw observational data and $-0.023 \pm 0.002$ ‰ (µmol mol$^{-1}$)$^{-1}$ for the corrected data. This rate of change for the corrected

data is less negative than approximately $-0.05$ ‰ (µmol mol$^{-1}$)$^{-1}$ for the seasonal cycles in the troposphere (Nakazawa et al., 1993) and is near the rate of change for their secular trends (Morimoto et al., 2000; Goto et al., 2017). Generally, because of the Brewer–Dobson circulation, air at higher altitudes is older. Therefore, we consider that this rate of change



for the corrected data  basically reflects  different ages of air.

### 3.4 Two-dimensional model of stratospheric $\delta^{13}CO_2$

The mechanisms governing the vertical $\delta^{13}CO_2$ profile in the stratosphere are schematically shown in Figure 6, including the influence of the age of air as well as the effects of $CH_4$ oxidation and GS discussed in Sects. 3.2 and 3.3. Tropospheric $\delta^{13}CO_2$ values have been decreasing through time due to the combustion of fossil fuels. If it is assumed that $\delta^{13}CO_2$ has no sink or source in the stratosphere, the air that intrudes from the tropical upper troposphere is slowly transported to the mid-latitude stratosphere by the Brewer–Dobson circulation. Therefore, based on the concept of age of air, it is expected that $\delta^{13}CO_2$ values should also decrease with time in the stratosphere after a certain lag time. Accordingly, $\delta^{13}CO_2$ values should be higher in the mid-stratosphere than in the troposphere (dotted line in Fig. 6). In contrast, GS decreases $\delta^{13}CO_2$ values at higher altitudes because $^{13}C^{16}O_2$ is heavier than $^{12}C^{16}O_2$ (dashed line in Fig. 6). In addition, $^{13}C$-depleted $CO_2$ produced by $CH_4$ oxidation further decreases $\delta^{13}CO_2$ values at high altitudes (solid line in Fig. 6).  The observed stratospheric vertical $\delta^{13}CO_2$ profile should therefore be formed by a combination of these effects.

To verify this hypothesis, we performed numerical simulations using a two-dimensional model of the middle atmosphere (SOCRATES) developed by the National Center for Atmospheric Research (NCAR; Huang et al., 1998; Park et al., 1999; Khosravi et al., 2002). Details of the model calculations are described in Appendix B. The monthly average meridional $\delta^{13}CO_2$ distribution calculated by SOCRATES for August 2016 is shown in Figure 7, which includes the effects of $CH_4$ oxidation and GS. The effect of the $CO_2$ production associated with the KIE on $CH_4$ destruction is included in a simplified manner (see Appendix B). The $\delta^{13}CO_2$ values decrease with increasing altitude up to about 14 km and then increase up to about 20 km in the mid-latitudes of the northern hemisphere during summer. The distribution of $\delta^{13}CO_2$ from the troposphere to the lower stratosphere should therefore be closely related to the propagation of its seasonal cycle in the troposphere. $\delta^{13}CO_2$ values reach a maximum at around 20 km and then gradually decrease. At high latitudes in the southern (winter) hemisphere, the vertical decrease of $\delta^{13}CO_2$ is significant, suggesting that the effects of $CH_4$ oxidation and GS are enhanced by the descending upper atmosphere.

To investigate the vertical $\delta^{13}CO_2$ profile in detail, we compared this model with our observations in Figure 2c. The modeled $\delta^{13}CO_2$ profile calculated without accounting for $CH_4$ oxidation and GS increased with increasing altitude, and it was 0.14 ‰ higher at altitudes around 35 km than near the tropopause. Because the average rate of change of the long-term $\delta^{13}CO_2$ record at MLO is $-0.027$ ‰ $yr^{-1}$ for the period 1990–2022, the lag time at 35 km (0.14 ‰ divided by 0.027 ‰ $yr^{-1}$) should be approximately 5.2 years, nearly consistent with the calculated mean age of air (see Appendix C). If the effect of GS is further taken into account, the $\delta^{13}CO_2$ decreased by approximately 0.07 ‰ at 35 km. The most significant impact on the modeled vertical profile was the effect of $CH_4$ oxidation, which imparted a decrease of 0.26 ‰ at around 35 km. The vertical profile calculated including all effects exhibited a maximum at an altitude of about 20 km and then decreased monotonically at greater altitudes, mainly because of the effect of $CH_4$ oxidation. However, the observed $\delta^{13}CO_2$ profile increased to about 25 km altitude, indicating that the model results clearly overestimated the $^{13}C$ depletion at higher altitudes. We note, though, that this model calculation included the seasonal cycle of tropospheric $\delta^{13}CO_2$ and nearly reproduced the vertical distribution observed at 14–20 km altitude, which is mainly due to the upward



propagation of the seasonal cycle from the upper troposphere to the lower stratosphere. Therefore, the difference between the observed and modeled results above 20 km was likely due mainly to an overestimation of the effect of $CH_4$ oxidation.

One possible reason for this difference was an overestimation of $CH_4$ loss at high altitudes. The observed $CH_4$ mole fraction at ~35 km altitude ranged between about 500 and 900 nmol mol$^{-1}$ (Fig. 4), but the model results indicated a monthly average of about 600 nmol mol$^{-1}$ for August, which was lower than the observed average value (see Fig. B1c in Appendix B). This discrepancy may be related to the fact that we weakened the mass stream function to reproduce a

realistic age of air (see Appendix B). Another reason may be that the model simulation set the KIE on $CH_4$ destruction in a simplified manner, with the apparent fractionation factors set uniformly at 0.9889 and 0.9866 for the lower and mid-stratosphere, respectively. In reality, the apparent fractionation factor for $CH_4$ oxidation varies with altitude and latitude, and it is determined not only by a complex process that depends on the paths of air masses, but also by the chemical reactions that occur along the pathway (e.g. Röckmann et al., 2011). In this study, the KIEs on the $CH_4$–$CO$–$CO_2$ reaction

processes were not calculated explicitly, but rather in a simplified manner, which may have resulted in an overestimation of the effect on $\delta^{13}CO_2$. However, the present model simulation was able to prove our hypothesis that three factors – $CH_4$ oxidation, GS, and air age – are essential to determining vertical $\delta^{13}CO_2$ profiles.

## 3.5 Stratospheric potential $\delta^{13}C$

From the above discussion, we considered that the total amount of $^{13}C$ in both $CO_2$ and $CH_4$, $\delta^{13}C_T$, was conserved. We defined $\delta^{13}C_T$ as:

$$\delta^{13}C_T = \frac{n(CO_2) \times \delta^{13}CO_2 + n(CH_4) \times \delta^{13}CH_4}{n(CO_2) + n(CH_4)} \ , \qquad (9)$$

which is quasi-conservative with respect to the KIEs associated with $CH_4$ oxidation and consequent $CO_2$ production in the stratosphere. Furthermore, as mentioned above, the effect of GS cannot be ignored in the stratosphere. Taking this

into account, we defined the stratospheric potential $\delta^{13}C$, $\delta^{13}C_P$, by correcting $\delta^{13}C_T$ for GS as:

$$\delta^{13}C_P = \delta^{13}C_T - \Delta_G(\delta^{13}CO_2) \ , \qquad (10)$$

where $\Delta_G(\delta^{13}CO_2)$ denotes the GS correction for $\delta^{13}CO_2$. Normally, $\Delta_G(\delta^{13}CO_2)$ gives a negative value, which means that it has a positive effect on $\delta^{13}C_P$. This equation does not include the effect of GS on $\delta^{13}CH_4$, but that contribution to $\delta^{13}C_P$ is negligibly small (about $10^{-4}$ ‰). Eqs. (9) and (10) were applied to the observed results to calculate stratospheric $\delta^{13}C_T$

and $\delta^{13}C_P$ profiles. For some air samples without $\delta^{13}CH_4$ measurements, values were estimated from the $CH_4$ mole fraction using a linear function fitted to the log-log plot of the observed data shown in Figure 4. The $\langle\delta_G\rangle$ values were also partially complemented by applying the average vertical profile. The average vertical profile of $\delta^{13}C_P$ thus calculated is shown in Figure 8, alongside the $CO_2$ mole fraction. The data plotted in this figure were detrended and normalized to August 2016. The $CO_2$ mole fractions were also corrected for $CH_4$ oxidation and GS, although the magnitude of these

corrections was relatively small. $\delta^{13}C_P$ increased with increasing altitude from the tropopause to about 25 km, then was almost constant with increasing altitude. Unlike $\delta^{13}CO_2$ (Fig. 2c), $\delta^{13}C_P$ was inversely correlated with the $CO_2$ mole fraction. Figure 9 shows the temporal variations of the averaged $\delta^{13}CO_2$, $\delta^{13}C_T$, and $\delta^{13}C_P$ values above 24 km altitude. $\delta^{13}C_T$ and $\delta^{13}C_P$ changed almost in parallel with $\delta^{13}CO_2$, showing a similar decreasing trend through time. The average difference from $\delta^{13}CO_2$ was $-0.079 \pm 0.025$ ‰ for $\delta^{13}C_T$ and $-0.034 \pm 0.027$ ‰ for $\delta^{13}C_P$.





To estimate the age of stratospheric air based on $\delta^{13}C_P$ observations in the stratosphere, $\delta^{13}C_P$ at the tropical upper troposphere was required. However, there are no long-term data records of $\delta^{13}C$ values in $CO_2$ and $CH_4$ in the tropical upper troposphere. Therefore, we hypothetically synthesized a $\delta^{13}C_P$ record in the tropical upper troposphere based on surface data observed at MLO and in the equatorial Pacific region by INSTAAR (White et al., 2024; Michel et al., 2023). In our previous study, we employed $CO_2$ mole fraction data obtained in the tropical upper troposphere using

the Automatic Air Sampling Equipment in the Comprehensive Observation Network for TRace gases by AIrLiner (CONTRAIL) program (Machida et al., 2008; Sawa et al., 2008; Matsueda et al., 2015) to estimate the mean age of stratospheric air (Sugawara et al., 2018). A similar method was used in this study. We first estimated the average seasonal cycle of $\delta^{13}CO_2$ in the tropical upper troposphere from the observed seasonal cycle of $CO_2$ mole fraction, assuming that the correlation between the seasonal cycles of $CO_2$ mole fraction and $\delta^{13}CO_2$ in the upper troposphere was approximately

$-0.05$ ‰ (µmol mol$^{-1}$)$^{-1}$ (Nakazawa et al., 1993). The long-term $\delta^{13}CO_2$ trend was calculated from the observation data at MLO and then adjusted to match the observations in the equatorial Pacific region. The average seasonal cycles were then added to the long-term $\delta^{13}CO_2$ trend estimated in the equatorial Pacific region. The difference between $\delta^{13}CO_2$ values at the surface and those in the upper troposphere in the equatorial region is currently unknown. However, Sugawara et al. (2018) have reported that the $CO_2$ and $SF_6$ ages in the TTL are 0.5–0.6 years, consistent with the $SF_6$ age of 0–1.5 years

in the TTL based on Michelson Interferometer for Passive Atmospheric Sounding (Stiller et al., 2008). Taking this into account, we assumed the difference between surface and upper troposphere $\delta^{13}CO_2$ values in the equatorial region to be equivalent to 0.5 years. Because systematic measurements of $CH_4$ mole fraction and $\delta^{13}CH_4$ in the upper troposphere were performed in the CONTRAIL program (Umezawa et al., 2012), the MLO data obtained by INSTAAR were adjusted to fit the CONTRAIL data in the equatorial upper troposphere to produce long-term records of $CH_4$ mole fraction and

$\delta^{13}CH_4$ in the equatorial upper troposphere. Because the seasonal cycle of $\delta^{13}CH_4$ in the equatorial upper troposphere is not clear (Umezawa et al., 2012), the seasonal cycle was ignored in the calculation of the upper tropospheric $\delta^{13}C_P$. Since GS, i.e. $<\delta_G> - <\delta_G>_0$, should be zero in the equatorial upper troposphere, tropospheric $\delta^{13}C_P = \delta^{13}C_T$.

          As seen in Figure 9, $\delta^{13}CO_2$ values did not differ significantly between the troposphere and the mid-stratosphere. However, the $\delta^{13}C_P$ values calculated from observed values in the mid-stratosphere were, on average, about

$0.14 \pm 0.03$ ‰ higher than those calculated for the tropical troposphere. This means that $\delta^{13}C_P$ is conservative in the stratosphere and that the decreasing tropospheric $\delta^{13}C_P$ trend through time propagates into the stratosphere with a time delay. Roughly calculated, this lag time is about 5 years, consistent with the age of air estimated from $CO_2$ mole fractions (Appendix C) and the model calculation. If the mid-stratospheric $\delta^{13}C_P$ values were shifted by $-5$ years, they were in good agreement with the tropospheric $\delta^{13}C_P$ trend (Fig. 9).

          It is thus possible to estimate the mean age of air using $\delta^{13}C_P$, which can serve as an additional air age tracer.

Therefore, we estimated the mean age of air not only from the $CO_2$ mole fraction but also from $\delta^{13}C_P$ using a convolution method (Ray et al., 2017; Sugawara et al., 2018) (described in Appendix C). It was necessary to consider that both the uncertainties on individual mole fraction or isotopic ratio measurements and those on the tropospheric reference record propagated to the uncertainties on individual age estimates (e.g. Leedham Elvidge et al., 2018). Regarding our $CO_2$ ages,

Umezawa et al. (2024) have estimated a total uncertainty of 0.72 years, and we estimated the uncertainties on $\delta^{13}C_P$ ages



in a similar manner: by adding normal pseudo-random numbers to all measured values and calculating $\delta^{13}C_P$ values using Eqs. (9) and (10). The uncertainty on individual $\delta^{13}C_P$ values was almost the same as that on $\delta^{13}C$, ±0.02 ‰. In contrast, the uncertainty on the tropospheric reference record for $\delta^{13}C_P$ was calculated to be ±0.03 ‰, which was the standard deviation when the curve fitting was applied to the MLO $\delta^{13}C$ data used as the tropospheric reference record. Taking into account all these uncertainties and the tropospheric trend of −0.027 ‰ yr$^{-1}$, the total uncertainty on individual $\delta^{13}C_P$ ages was estimated to be ±1.9 years. This rather large uncertainty means that $\delta^{13}C_P$, with its smaller tropospheric secular change compared to other age tracers, was somewhat disadvantageous for age estimations. Because the seasonal cycle of $\delta^{13}C_P$ is significantly large in the troposphere, its signal propagates into the TTL and then into the stratosphere via the slow air transport. Simultaneously, atmospheric mixing broadens the width of the age spectrum, attenuating seasonal amplitudes with increasing altitude. Because some of the $\delta^{13}C_P$ values in the lower stratosphere showed high variability, with values lower than the tropospheric reference record, the convolution method could not provide reasonable age estimations for those data. We attributed these irregularities and low values to the large seasonal variations in the upper troposphere and the stratosphere–troposphere air exchange in the lower stratosphere. Therefore, only mid-stratospheric (>24 km altitude) data were taken into account for age estimations, and the average values of the mean age were calculated.

The results are shown in Figure 10, along with the mean age calculated from the $CO_2$ mole fraction using the same method (see Appendix C). The average $\delta^{13}C_P$ and $CO_2$ ages were 5.5 ± 1.6 and 4.4 ± 0.6 years, respectively. Because the uncertainty of the $\delta^{13}C_P$ age was large, no significant difference was apparent between the two values. The ratio of moments and its influence on age estimates have been examined in previous studies (e.g. Hauck et al., 2019; Nguyen et al., 2021). Here, as a sensitivity test, the mean age calculation was repeated for different ratios of moments ranging from 0.05 to 2.00 years. As a result, the mean ages derived from $\delta^{13}C_P$ were found to be within the observational uncertainties on the $\delta^{13}C_P$ age. To examine the long-term trend of mean age, a linear function was fitted to both the $CO_2$ and $\delta^{13}C_P$ age data using the least squares method. A slightly positive trend (0.04 ± 0.01 years yr$^{-1}$) was observed for the $CO_2$ age, but no specific trend was observed for the $\delta^{13}C_P$ age. It is important to note that our data were unevenly distributed over time, with sparse observations in recent years and more frequent observations before 2007. This made it difficult to clarify the exact 35-year trend from our data alone. However, the result that $\delta^{13}C_P$ age and $CO_2$ age are roughly consistent implies that $\delta^{13}C_P$ age could be used in addition to $SF_6$, halocarbons, etc., for better multi-component age estimations (Umezawa et al., 2024).

## 4 Conclusions

Previous studies on $\delta^{13}CO_2$ have mainly focused on observations in the troposphere and on elucidating the global $CO_2$ budget based on monitoring programs. However, little research has been performed on $\delta^{13}CO_2$ in the stratosphere, and the mechanisms of its distribution and variations are not fully understood. This study presented a comprehensive analysis of stratospheric $\delta^{13}CO_2$ obtained from balloon observations over Japan during the period 1985–2020. The following four factors were found to be important for understanding the variations of $\delta^{13}CO_2$ in the stratosphere. (1) Appropriate corrections for the MIE on $^{17}O$ and $^{18}O$ in isotopic analyses are essential. However, even after applying this correction, we found that the average values of $\delta^{13}CO_2$ in the northern mid-latitude mid-stratosphere and the tropical troposphere



were almost identical, despite the secular decreasing $\delta^{13}CO_2$ trend and the expected time lag (age of air). We attributed this strange agreement to the other three factors: (2) airborne production of $^{13}C$-depleted $CO_2$ by the oxidation of $CH_4$ in the stratosphere, (3) GS of $^{12}CO_2$ and $^{13}CO_2$ in the stratosphere, and (4) propagation of the decreasing tropospheric $\delta^{13}CO_2$

trend into the stratosphere. Based on these considerations, we introduced 'stratospheric potential $\delta^{13}C$' ($\delta^{13}C_P$) as a new concept and inspected $\delta^{13}C_P$ in terms of the age of air in detail. Mean age of air was estimated by using $\delta^{13}C_P$ value with the convolution method. The average $\delta^{13}C_P$ age was calculated to be $5.5 \pm 1.6$ years for the period of 1985-2020. Although the $\delta^{13}C_P$ age has larger uncertainties than the $CO_2$ age at present, our results showed that stratospheric $\delta^{13}C_P$ can serve as an additional age tracer alongside $CO_2$ and $SF_6$ mole fractions, and it should prove a useful tool for investigating

stratospheric transport processes.

        $\delta^{13}C_P$ age is slightly larger than $CO_2$ age ($4.4 \pm 0.6$ years) by about 1.1 years on average. In this regard, Engel et al. (2009) reported that $CO_2$ age and $SF_6$ age observed in the mid-stratosphere at mid-latitudes during the period 1975-2005 ranged from 4.7 to 5.3 years. Compared to their reported ages, our $\delta^{13}C_P$ age and $CO_2$ age tend to be slightly larger and smaller, respectively. It is likely that $\delta^{13}C_P$ age is overestimated because the effect of methane oxidation was

underestimated in the calculation of $\delta^{13}C_T$. The discrepancy between the two $CO_2$ ages would be due to differences in the age estimation methods, the tropospheric reference records, and the ratios of moments used in the two studies. Our GS correction for the $CO_2$ mole fraction also has the effect of lowering $CO_2$ age. Therefore, it is necessary to perform age estimations by analyzing all observational data using the same method in future study. It is also needed to reduce uncertainty in age estimation by performing multi-component age estimations using additional age tracers, such as

halocarbons and $\delta^{13}C_P$.

        The observation of numerous stratospheric air constituents, as in this study, requires the collection of a large amount of stratospheric air with a high-quality method and multi-component gas analyses of each air sample. GS of atmospheric constituents in the stratosphere was first revealed by our previous balloon observations, and this knowledge is indispensable for a better understanding of the variations of stratospheric $\delta^{13}CO_2$. The quality of air sampling is

particularly important for observing GS in the mid-stratosphere, because even slight separations due to molecular diffusion during air sampling could prevent GS observations. At present, the only method capable of such sampling is our cryogenic air sampler.

        The 2-D model reproduced the basic structure of stratospheric $\delta^{13}CO_2$ and elucidated the mechanisms governing $\delta^{13}CO_2$ behavior. However, our numerical model simulations remained insufficient because the mean age of

stratospheric air could not be reproduced adequately without arbitrarily tuning the model. Accordingly, our model overestimated the effect of airborne $CO_2$ production via $CH_4$ oxidation because the reaction process was treated in a very coarse manner. To improve upon this model, it will be necessary to incorporate KIEs into all reactions related to stratospheric carbon. Although a two-dimensional model was used in this proof-of-concept study, a three-dimensional model is desirable for more realistic simulations in the future.


**Appendix A: Gravitational separation of major atmospheric components**



The gravitational separation (GS) of atmospheric components in the stratosphere was first reported from our balloon observations in Japan (Ishidoya et al., 2006, 2008a, 2013), in which the isotopic and elemental ratios of major atmospheric

compositions, such as $\delta^{15}N$ in $N_2$, $\delta^{18}O$ in $O_2$, and $\delta(Ar/N_2)$, were measured by mass spectrometry at the National Institute of Advanced Industrial Science and Technology (AIST) (e.g. Ishidoya et al., 2006, 2008a; Sugawara et al., 2018). Technical aspects of our mass spectrometry analyses are described in detail in Ishidoya and Murayama (2014). In this study, we used $\delta^{15}N$, $\delta^{18}O$, and $\delta(Ar/N_2)$ to evaluate GS, which are defined as:

$$\delta^{15}N = \frac{[n(^{15}N^{14}N)/n(^{14}N^{14}N)]_{sp}}{[n(^{15}N^{14}N)/n(^{14}N^{14}N)]_{st}} - 1, \quad \text{(A1a)}$$

$$\delta^{18}O = \frac{[n(^{18}O^{16}O)/n(^{16}O^{16}O)]_{sp}}{[n(^{18}O^{16}O)/n(^{16}O^{16}O)]_{st}} - 1, \quad \text{(A1b)}$$

and

$$\delta(Ar/N_2) = \frac{[n(^{40}Ar)/n(^{14}N^{14}N)]_{sp}}{[n(^{40}Ar)/n(^{14}N^{14}N)]_{st}} - 1, \quad \text{(A1c)}$$

where 'sp' and 'st' denote the sample and standard gases, respectively. The $1\sigma$ reproducibilities of our $\delta^{15}N$, $\delta^{18}O$, and $\delta(Ar/N_2)$ measurements were $\pm 2$, $\pm 5$, and $\pm 7$ per meg, respectively, which was sufficient to detect GS in the stratosphere.

Ishidoya et al. (2013) have concluded that vertical variations of the isotopic ratios of major atmospheric components are dominated by differences in molecular mass, based on mass-dependent relationships among related molecules such as $\delta^{15}N$ in $N_2$, $\delta^{18}O$ in $O_2$, and the $Ar/N_2$ ratio. Such GS in the stratosphere is known to be proportional to the mass difference, as would be expected for pure molecular diffusion, even though eddy diffusion obviously far exceeds molecular diffusion in the stratosphere. Therefore, GS can be expressed as follows:

$$\delta - \delta_0 = \Delta m \times (\langle\delta_G\rangle - \langle\delta_G\rangle_0), \quad \text{(A2)}$$

where $\Delta m$ is the mass number difference, the suffix "0" refers to the value before GS occurs (i.e. corresponding to the relevant value in the upper troposphere), and $\langle\delta_G\rangle$ is the average of $\delta$ values normalized to $\Delta m = 1$ as reported by Ishidoya et al. (2013) and Sugawara et al. (2018). $\langle\delta_G\rangle$ is defined by:

$$\langle\delta_G\rangle = \frac{1}{3}\left[\delta^{15}N + \delta^{18}O/2 + \delta\left(Ar/N_2\right)/12\right]. \quad \text{(A3)}$$

Figure A1 shows vertical profiles of $\langle\delta_G\rangle$ obtained from balloon observations over Japan (Ishidoya et al., 2013). The value of $\langle\delta_G\rangle$ decreases gradually with increasing altitude. By fitting a linear function to the vertical profiles, the average difference of $\langle\delta_G\rangle$ between the lowermost and uppermost altitudes was found to be $-66 \pm 17$ per meg. Here, it important to mention the relationship between the GS of $\delta^{18}O$ and photochemical processes. As mentioned in Sect. 2.2., in the stratosphere, $O_3$ and $CO_2$ are significantly enriched in $^{17}O$ and $^{18}O$ by MIEs associated with photochemical reaction

processes. Because stratospheric $O_2$ molecules are the source of the $^{17}O$ and $^{18}O$ enrichment, $\delta^{17}O$ and $\delta^{18}O$ values of $O_2$ are thought to be lower in the stratosphere than in the troposphere due to photochemical processes. This assumption is crucial not only for the Dole–Morita effect, but also for the $^{17}O$ budget of tropospheric $O_2$ (Bender et al., 1994; Luz et al., 1999; Luz and Barkan, 2011; Ishidoya et al., 2025). However, $\delta^{18}O$ values of stratospheric $O_2$ measured in our balloon observations show a very large decrease, i.e. by approximately $-130$ per meg, at around 35 km that is almost entirely

dominated by GS. This decrease is probably also due in part to photochemical effects, but their contribution should be



negligible, amounting to only a few per meg (Ishidoya et al., 2025).

**Appendix B: Two-dimensional model**

We used a two-dimensional model of the middle atmosphere (SOCRATES) developed by the National Center for

Atmospheric Research (NCAR) (Huang et al., 1998; Park et al., 1999; Khosravi et al., 2002). The advantage of using this

model is that we have already calculated $^{13}C^{16}O_2$ and $^{12}C^{16}O_2$ and examined stratospheric GS in our previous studies

(Ishidoya et al., 2013; Sugawara et al., 2018). Therefore, we give only a brief description of the model calculation here.

To reproduce GS, the molecular diffusion flux must be calculated. The vertical component of the molecular diffusion flux

($F_{m_i,z}$) is given by:

$$F_{m_i,z} = -D_{m_i}\left\{\frac{\partial n_i}{\partial z} + \frac{m_i g}{RT}n_i + \left(1 + \alpha_{T_i}\right)\frac{\partial(\ln T)}{\partial z}n_i\right\} \quad , \quad \text{(B1)}$$

where $n_i$, $D_{m_i}$, $m_i$, and $\alpha_{T_i}$ are the number density, molecular diffusion coefficient, molecular mass, and thermal diffusion

factor of atmospheric constituent $i$, respectively, and $g$, $R$, and $T$ are the gravitational acceleration, gas constant and

temperature, respectively (Banks and Kockarts, 1973). However, because the thermal diffusion flux is not important in

this case, we ignored it in the model calculations. Because it is cumbersome to include all $CO_2$ isotopologues in the model

calculations, only $^{12}C^{16}O_2$ and $^{13}C^{16}O_2$ were calculated independently in the model, and $\delta^{13}CO_2$ was approximated using

the respective mole fractions of $n(^{13}C^{16}O_2)$ and $n(^{12}C^{16}O_2)$ as:

$$\delta^{13}CO_2 = \left(\frac{1}{^{13}R_{VPDB}}\frac{n(^{13}C^{16}O_2)}{n(^{12}C^{16}O_2)} - 1\right) . \quad \text{(B2)}$$

Because $\delta^{13}CO_2$ has a large seasonal cycle in the troposphere, we used NOAA/ESRL data to give latitude-dependent

seasonal cycles as boundary conditions at the model surface. In addition, a secular decrease of $\delta^{13}CO_2$ in the troposphere

was given by the MLO data (White et al., 2024). SOCRATES originally included a series of carbon reactions starting

with $CH_4$ oxidation. Model calculations were repeated with and without GS, the secular trend and seasonal cycle of

tropospheric $\delta^{13}CO_2$, and airborne $CO_2$ production, and comparisons were made between them (Fig. 2c). The chemical

budget of $CO_2$ in SOCRATES is calculated as:

$$\frac{d[CO_2]}{dt} = k_{CO+OH}[CO][OH] + k_{CO+O(^3P)}[M][CO][O(^3P)] - J_{CO_2}[CO_2] \quad , \quad \text{(B3)}$$

where $k_{CO+OH}$ and $k_{CO+O(^3P)}$ are the reaction rates of their respective chemical reactions, and $M$ is the third body which

carries off excess energy in a termolecular reaction. Because $CO_2$ photodissociation is important only above the

mesosphere (e.g. Garcia et al., 2014), it was ignored in this study. KIEs in the $CH_4$–$CO$–$CO_2$ chain reaction were not

included explicitly in these simulations but were rather treated in a coarse manner as follows. Simulations were performed

with and without the chemical production of $CO_2$. Taking the $CO_2$ mole fractions simulated with ('wP') and without

chemical production ('nP') to be $n(CO_2)_{wP}$ and $n(CO_2)_{nP}$, respectively, the $CO_2$ mole fraction difference between the two

simulations, $n(CO_2)_{wP} - n(CO_2)_{nP}$, was calculated as the apparent $CO_2$ production in the stratosphere. The apparent $CO_2$

production is distinct from the local production calculated in Eq. (B3) because it is an apparent value influenced by the

accumulated chemical processes along the pathway of the air mass, including transport and mixing. A typical value of the

apparent $CO_2$ production was simulated to be about 1.5 μmol mol$^{-1}$ at 35 km altitude in the mid-latitudes. Similarly, the

$\delta^{13}CO_2$ value simulated excluding chemical $CO_2$ production was expressed by $\delta^{13}CO_{2,nP}$. After both these simulations, the



$\delta^{13}CO_2$ value including airborne $CO_2$ sources from the $CH_4$ destruction, $\delta^{13}CO_{2,wP}$, was calculated as:

$$\delta^{13}CO_{2,wP} = \frac{\delta^{13}CO_{2,nP} \times n(CO_2)_{nP} + \delta^{13}CH_{4\_L} \times [n(CO_2)_{wP} - n(CO_2)_{nP}]}{n(CO_2)_{wP}} . \quad \text{(B4)}$$

In this equation, $\delta^{13}CH_{4\_L}$ was calculated using Eq. (6) assuming apparent fractionation factors of 0.9889 and 0.9866 at
lower (<24 km) and upper (>24 km) altitudes, respectively (see Sect. 3.2). Note that $\delta^{13}CH_{4\_L}$ values were estimated

from the apparent fractionation factors by assuming Rayleigh distillation relationships between vertical profiles of $CH_4$
mole fraction and $\delta^{13}CH_4$. Therefore, these apparent fractionation factors differed from fractionation factors sensu stricto
and were influenced by the accumulated KIEs along the pathway of the air mass, as with the apparent $CO_2$ production
described above. Eq. (B4) means that $\delta^{13}CO_2$ was simulated by combining the apparent fractionation factors estimated
from observations with the apparent $CO_2$ production simulated by the model.

A virtual clock tracer was used to calculate the mean age of air. However, previous research has shown that
the speed of the Brewer–Dobson circulation in SOCRATES is fast, leading to an underestimation of age of air. Therefore,
similar to Sugawara et al. (2018), we reduced the mass stream function to approximate a realistic mean age, which also
improved the reproduction of GS. The model results of GS reproduced well the almost linear decrease with increasing
altitude above the tropopause in the observational results (Fig. A1). The model calculations were carried out over the 25

years from 1995 to 2020. The monthly average meridional distribution of the mean air age, GS, and $CH_4$ mole fraction
calculated by SOCRATES for August 2016 is shown in Figure B1.

**Appendix C: $CO_2$ mole fraction and age calculations**

Corresponding to the secular decrease of $\delta^{13}CO_2$, an increase in $CO_2$ mole fraction was also clearly observed in the mid-

stratosphere. To compare the temporal variations of the $CO_2$ mole fraction in the mid-stratosphere with those in the
troposphere, the average values above 24 km altitude were calculated for each year, shown in Figure C1 alongside the
monthly average data obtained by NOAA/ESRL at MLO. The results of the WMO/IAEA Round Robin Comparison
Experiment showed that our $CO_2$ mole fraction values were approximately 0.2 μmol mol$^{-1}$ higher than the NOAA/ESRL
values; accordingly, our $CO_2$ mole fraction values are plotted in Figure C1 only after subtracting 0.2 μmol mol$^{-1}$. The

$CO_2$ mole fraction in the mid-stratosphere over Japan increased monotonously and reached about 400 μmol mol$^{-1}$ in 2020,
lagging behind that at MLO by about 5 years. The average rate of change of the $CO_2$ mole fraction in the mid-stratosphere
was calculated to be $1.68 \pm 0.04$ μmol mol$^{-1}$ yr$^{-1}$ by applying the least-squares method to the observed values. However,
this increasing trend was not linear, gradually steepening over the last 40 years. Therefore, a quadratic function of the
form $n(CO_2) = K_0 + (K_1 \times t) + (K_2 \times t^2)$ would better represent the observed variations. Here, $t$ (years) is the elapsed time

since 1980. The coefficients of $CO_2$ mole fraction in the mid-stratosphere $K_0$, $K_1$, and $K_2$ were calculated to be 333.7,
1.0448, and 0.014755, respectively. The standard deviation of residuals was 0.8 μmol mol$^{-1}$. The same procedure was
applied to the annual mean $CO_2$ mole fractions at MLO for the period 1980–2015, considering a time lag of 5 years
between the mid-stratosphere and the troposphere. The coefficients $K_0$, $K_1$, and $K_2$ for the MLO data were calculated to
be 338.85, 1.2307, and 0.014577, respectively. As seen in Figure C1, the two quadratic functions thusly obtained agreed

well to within $\pm 0.7$ μmol mol$^{-1}$. This result suggested that even temporal changes in the rate of increase in the troposphere



propagate to the mid-stratosphere with a certain time delay.

The $CO_2$ age based on our balloon observations has been partly reported in previous studies (Engel et al., 2002; Ishidoya et al., 2013; Umezawa et al., 2024). $CO_2$ ages estimated using air samples collected from 1995 to 2010 have been published as supplementary data in Ishidoya et al. (2013). However, the method of age estimation was

subsequently updated in Sugawara et al. (2018). In this study, we further improved the estimation method by correcting the $CO_2$ mole fraction data for GS and created a new 35-year $CO_2$ age record. We estimated the mean age of air from the $CO_2$ mole fraction using the convolution method (Ray et al., 2017; Leedham Elvidge et al., 2018; Sugawara et al., 2018); this method has been described in detail by Sugawara et al. (2018), so only a brief explanation is given here. Hypothetical age spectra were used in the convolution method to estimate the mean age (Waugh and Hall, 2002). Expected temporal

variations in the $CO_2$ mole fraction in the stratosphere, $x(\Gamma, t)$, were calculated by convolution of the tropospheric reference curve, $x_0(t)$, and the age spectrum, $G(\Gamma, t)$:

$$x(\Gamma, t) = \int_{t-T_B}^{t} x_0(t') G(\Gamma, t - t') dt', \qquad \text{(C1)}$$

where $T_B$ is the integration time interval. $T_B$ should theoretically be $\infty$, but we truncated it to 20 years for the calculations. Actual age spectra, $G(\Gamma, t)$ are usually unknown. Therefore, we used the inverse Gaussian distribution (Waugh and Hall,

2002) as:

$$G(\Gamma, t) = \left(\frac{\Gamma^3}{4\pi\Delta^2 t^3}\right)^{1/2} \exp\left[\frac{-\Gamma(t-\Gamma)^2}{4\Delta^2 t}\right], \qquad \text{(C2)}$$

where $\Delta$ denotes the width of the age spectrum, which was parameterized using the mean age ($\Gamma$), i.e. the ratio of moments, $\Delta^2/\Gamma$. This value has usually been assumed to be 0.7 years, as suggested by Hall and Plumb (1994) based on results from a General Circulation Model calculation, and was used to estimate $SF_6$- and $CO_2$-derived mean ages in the northern mid-

and high-latitude stratosphere (Engel et al., 2002). Recently, however, Fritsch et al. (2020) have reported that a value of 1.25 years is better for estimating ages from the $SF_6$ mole fraction. Therefore, we used 1.25 years for the ratio of moments in this study. After calculating the convolutions, the $CO_2$ age was determined by substituting the observed $CO_2$ mole fractions, $x_{obs}$, into the inverse function, $\Gamma(x, t)$.

As described in the main text, the influences of $CO_2$ production by $CH_4$ oxidation and GS should be

considered to precisely estimate the $CO_2$ age. Therefore, the $CO_2$ mole fraction was corrected as follows. At first, the tropical upper tropospheric $CH_4$ data, $x_{CH_4\_tp}(t)$, were created by adjusting the annual mean data at MLO to fit the CONTRAIL data in the tropical upper troposphere (Umezawa et al., 2012). Then, we corrected $x_{obs}$ as:

$$x_{cor}(t) = x_{obs}(t) - \left[x_{CH_4\_tp}(t - \Gamma) - x_{CH_4\_obs}(t)\right] - \Delta_G(x_{obs}(t)), \qquad \text{(C3)}$$

where $x_{cor}$ and $x_{CH_4\_obs}$ denote the corrected $CO_2$ mole fraction and observed $CH_4$ mole fraction, respectively. $\Delta_G$ is the

correction for GS (see Sect. 3.3.). The $CO_2$ mole fractions corrected for $CH_4$ oxidation and GS are shown with the convolutions in Figure C2. Finally, the $CO_2$ age was determined as $\Gamma_{CO_2} = \Gamma(x_{cor}, t)$. We note that Eq. (C3) contains the mean age of air, $\Gamma$, because the $CH_4$ mole fraction in the troposphere has increased with time and $CH_4$ is destroyed over the air transport pathway from the tropical upper troposphere to the observation altitude in the stratosphere. Therefore, $\Gamma_{CO_2}$ was solved iteratively by starting from $\Gamma_{CO_2} = 0$. This iteration converged sufficiently after the second time of $\Gamma_{CO_2}$

calculation. The vertical profiles of $\Gamma_{CO_2}$, corresponding to $CO_2$ age, are shown in Figure C3. The $\delta^{13}C_P$ age was also





calculated using the same method, except that the corrections for $CH_4$ oxidation and GS were already included when calculating $\delta^{13}C_P$. The convolutions are shown with the calculated $\delta^{13}C_P$ values in Figure C4.

**Data availability.** The observational data obtained by our balloon measurements are included as an electronic supplement to the manuscript.

**Author contributions.** SS designed the study, conducted the measurements of mole fractions of greenhouse gases, and drafted the manuscript. SM, TN, SA, and SS conducted the measurements of carbon isotopic ratios of $CO_2$. SM, SS, and
TU conducted the measurements of carbon isotopic ratios of $CH_4$. SI conducted the measurements of isotopic and elemental ratios of atmospheric major compositions. KI conducted the measurements of $N_2O$ mole fraction. HH, TN, SA, SM, and SS conducted the balloon experiments. SI, ST, and DG participated in balloon experiments. All authors approved the final manuscript.

**Competing interests.** The corresponding author declares that none of the authors has any competing interests.

**Acknowledgements.** We deeply thank the Scientific Ballooning (DAIKIKYU) Research and Operation Group of the Institute of Space and Astronautical Science (ISAS), JAXA, Japan.

**Financial support.** This study was supported by Japan Society for the Promotion of Science KAKENHI grants (grant nos. 22H05006, 23H00513, and 23K11396).

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

**Figures**

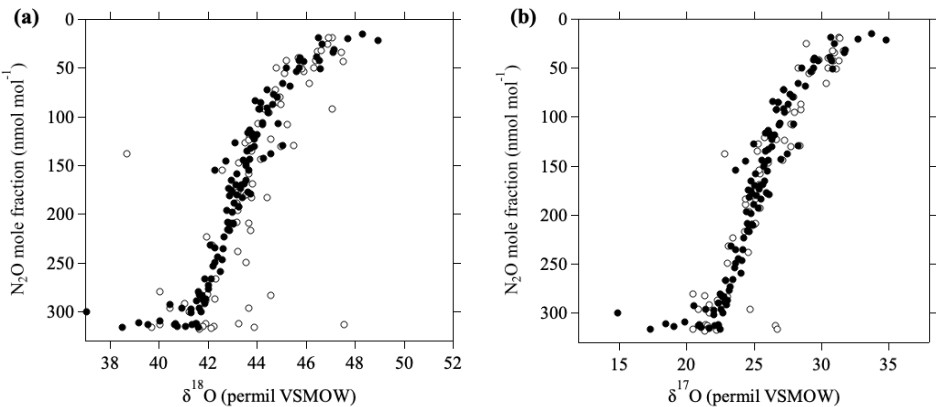


**Figure 1.** The relationship between the $N_2O$ mole fraction and (**a**) $\delta^{18}O$ in $CO_2$ and (**b**) $\delta^{17}O$ in $CO_2$. $\delta^{18}O$ and $\delta^{17}O$ values

measured by Kawagucci et al. (2008) are shown by open circles. $\delta^{18}O$ and $\delta^{17}O$ values calculated assuming a MIE factor

of 1.7 (see text) in this study are shown by closed circles.





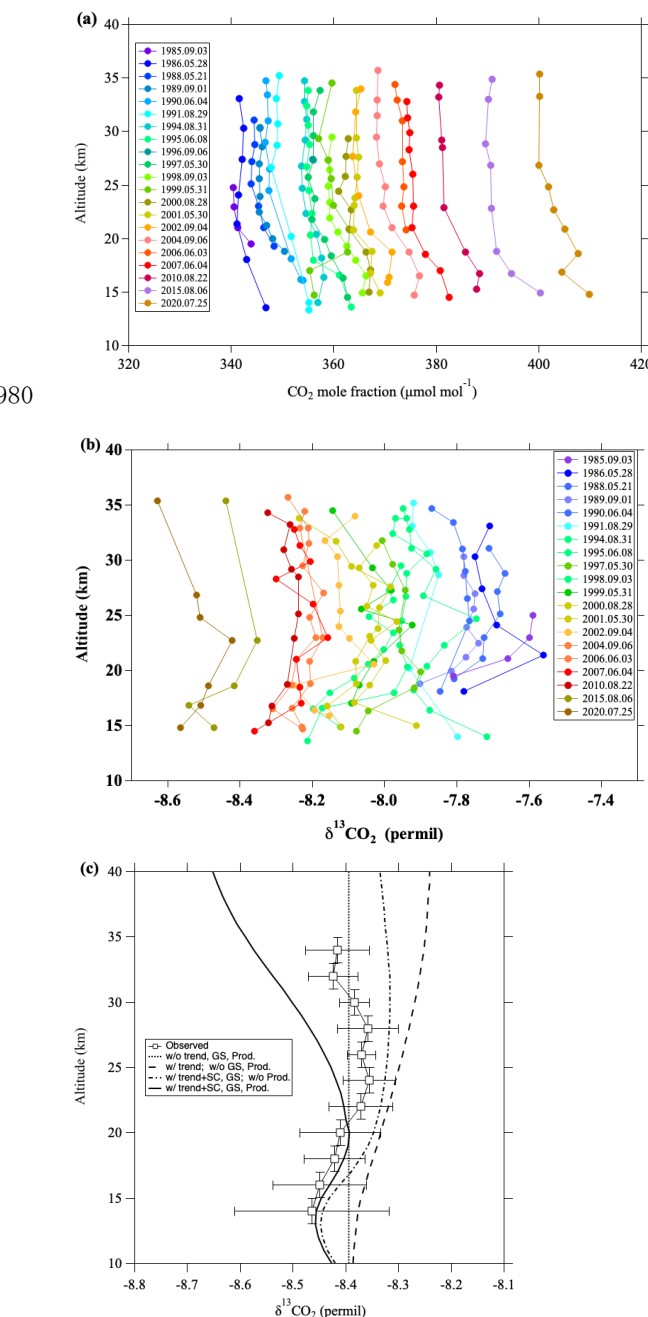


**Figure 2.** Vertical profiles of (**a**) $CO_2$ mole fraction and (**b**) $\delta^{13}CO_2$ over Japan during the period 1985–2020. (**c**) $\delta^{13}CO_2$

values averaged over the study period and grouped in eleven 2-km vertical bins (open squares). Curves show 2-D

modeling results without the tropospheric trend, GS, or airborne sources (dotted line), including only the tropospheric

trend (dashed line), including the tropospheric trend, seasonal cycle, and GS (dotted-dashed line), and including the

tropospheric trend, seasonal cycle, GS, and airborne sources (solid line).



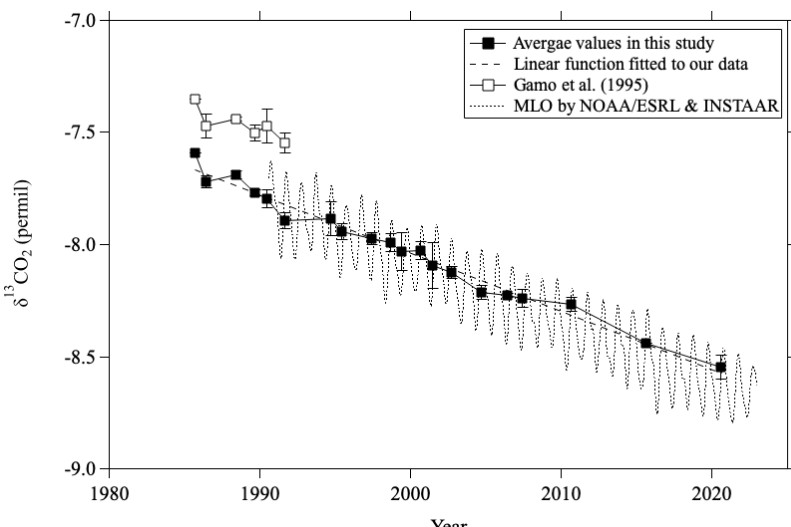

**Figure 3.** Average mid-stratospheric (>24 km altitude) $\delta^{13}CO_2$ values over Japan (closed squares). The linear function fitted to the average values using the least-squares method is shown by the dashed line. Values reported by Gamo et al. (1995) are shown by open squares. The best-fit curve for $\delta^{13}CO_2$ observed at Mauna Loa by NOAA/ESRL and INSTAAR is also shown by the dotted line.




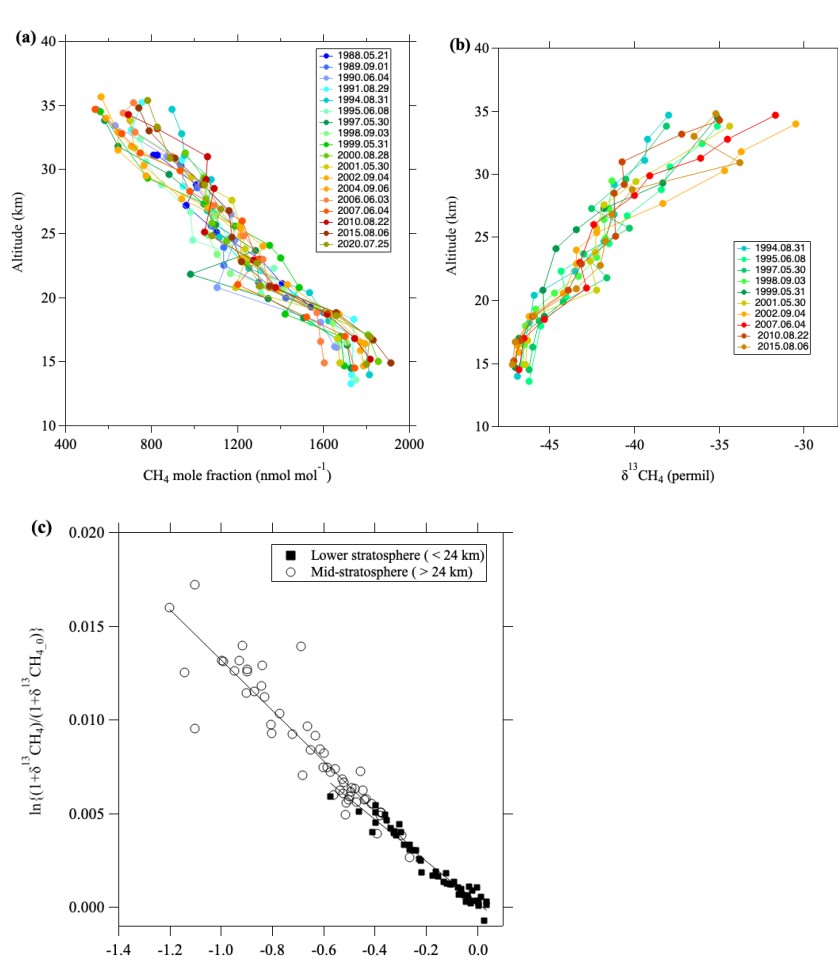

**Figure 4.** Vertical profiles of (**a**) $CH_4$ mole fraction and (**b**) $\delta^{13}CH_4$ over Japan, and (**c**) the relationship between $\ln\{(\delta^{13}CH_4 + 1)/(\delta^{13}CH_{4\_0} + 1)\}$ and $\ln([CH_4]/[CH_4]_0)$. In (**c**), the data are divided into the lower (<24 km, closed squares) and mid-stratosphere (>24 km, open circles), and the linear functions fitted to each dataset using the least-squares method are shown by lines.






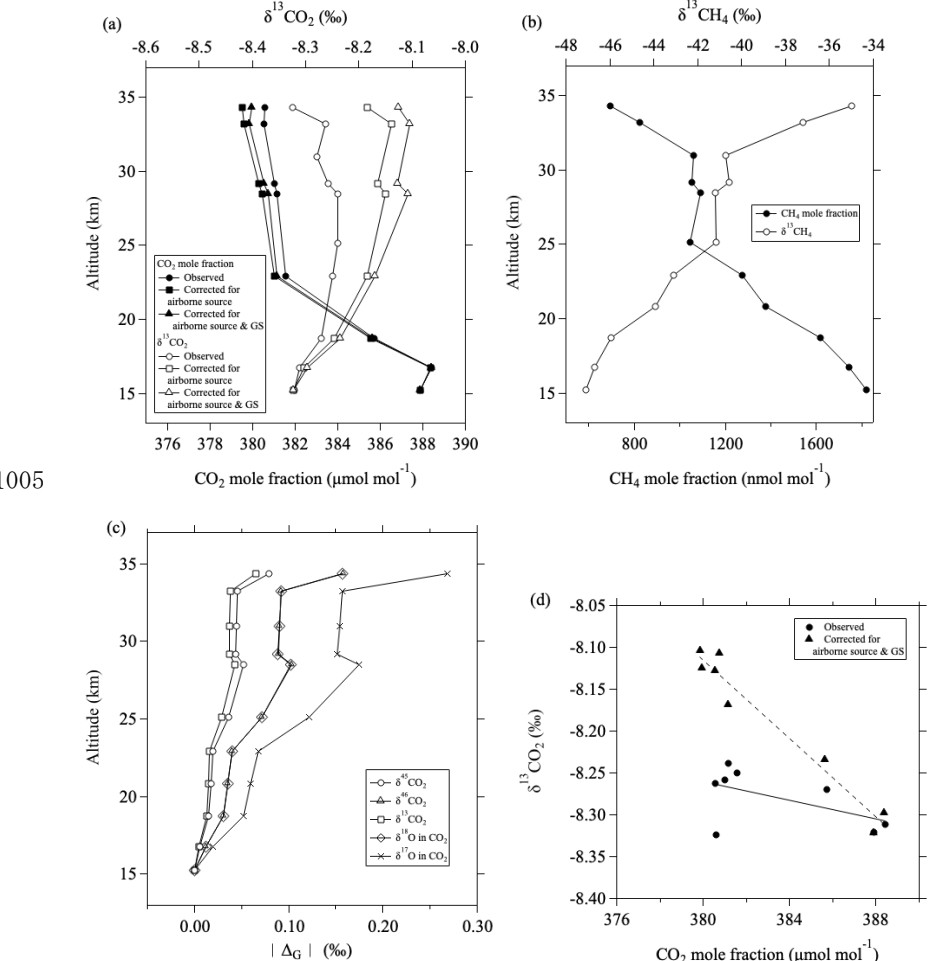


**Figure 5. (a)** Vertical profiles of the $CO_2$ mole fraction (closed circles) and $\delta^{13}CO_2$ (open circles) observed over Japan on 22 August 2010. Values corrected for airborne $CO_2$ sources (squares) and additionally for GS (triangles) are also shown. **(b)** Vertical profiles of the $CH_4$ mole fraction (closed circles) and $\delta^{13}CH_4$ (open circles). **(c)** Vertical profiles of the

magnitude of GS corrections, $|\Delta_G|$, for $\delta^{45}CO_2$ (circles), $\delta^{46}CO_2$ (triangles), $\delta^{13}CO_2$ (squares), and $\delta^{18}O$ (diamonds) and $\delta^{17}O$ (crosses) in $CO_2$. **(d)** Observed $\delta^{13}CO_2$ plotted against $CO_2$ mole fraction (closed circles) and those corrected for airborne $CO_2$ sources and GS (triangles). Lines are linear functions fitted to each trend by the least-squares method.





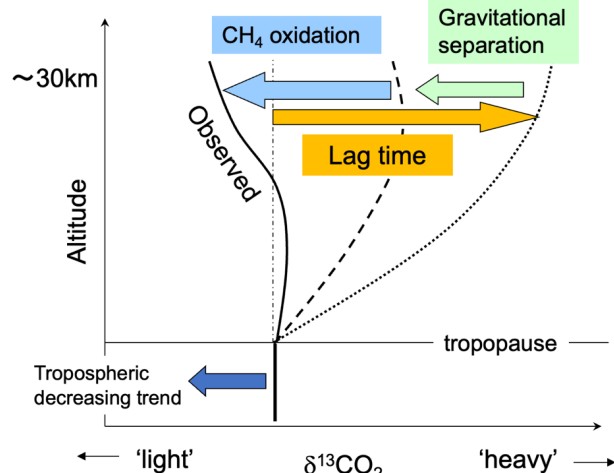


**Figure 6.** Schematic representation of the mechanisms influencing the vertical profile of stratospheric $\delta^{13}CO_2$ (see text).

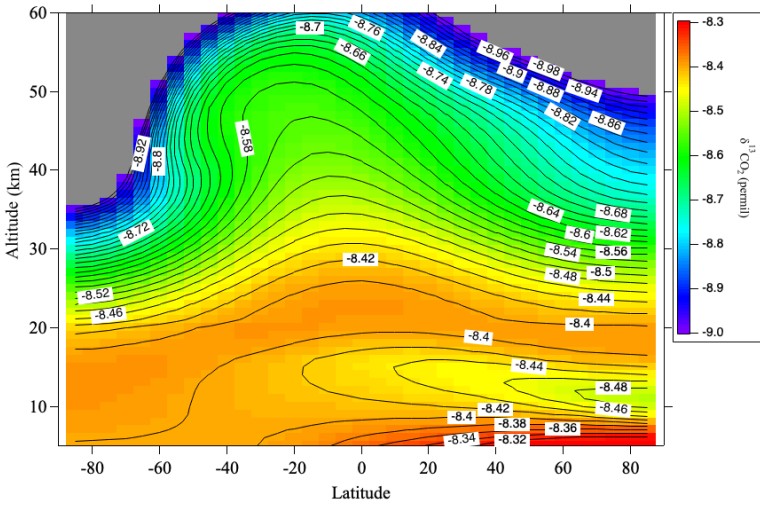

**Figure 7.** Monthly average meridional distribution of $\delta^{13}CO_2$ for August 2016 calculated using the SOCRATES model.

Values lower than −9.0 ‰ are colored gray.





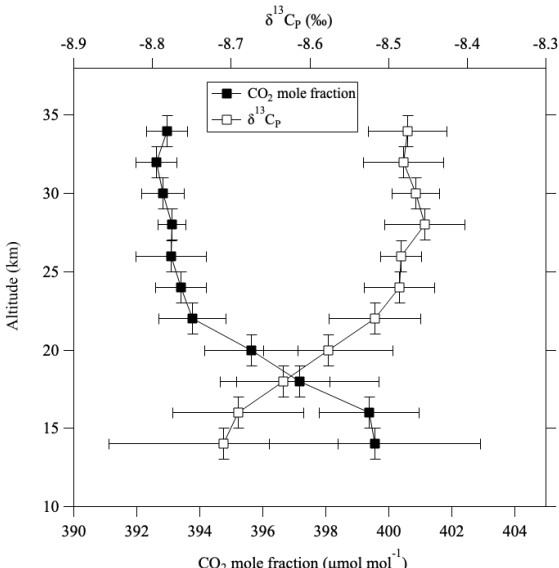

**Figure 8.** Vertical profiles of $CO_2$ mole fraction (closed squares) and stratospheric potential $\delta^{13}C$ ($\delta^{13}C_P$, open squares) based on their averages in 11 vertical bins over the period 1985–2020. All values were detrended and normalized to values in August 2016.

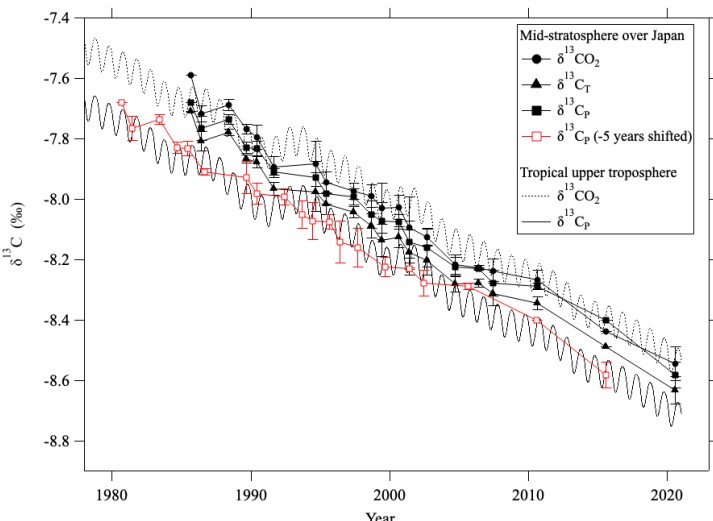

**Figure 9.** Average mid-stratospheric (>24 km altitude) values of $\delta^{13}CO_2$ (closed circles), $\delta^{13}C_T$ (closed triangles), and $\delta^{13}C_P$ (closed squares) over Japan. $\delta^{13}CO_2$ and $\delta^{13}C_P$ in the tropical upper troposphere are shown by dashed and solid lines, respectively. Red squares show the mid-stratospheric $\delta^{13}C_P$ shifted by −5.0 years.





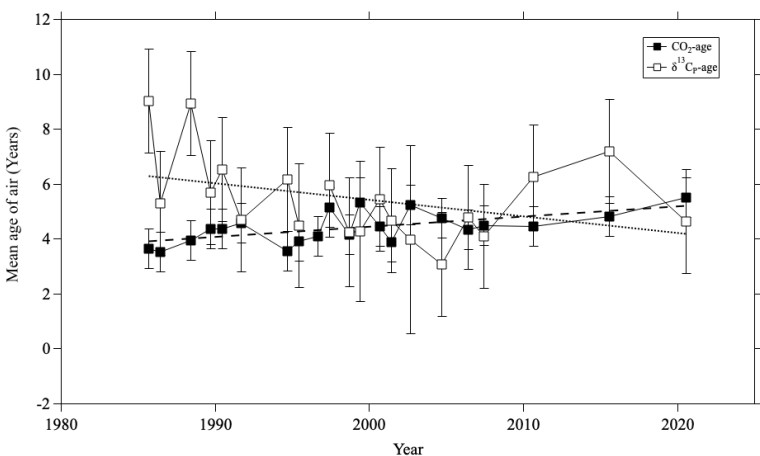

**Figure 10.** Temporal variations of averaged $\delta^{13}C_P$ ages (open squares) and $CO_2$ ages (closed squares) in the mid-
stratosphere (>24 km altitude) over Japan. Dotted and dashed lines are linear least-squares fits to the $\delta^{13}C_P$ and $CO_2$ ages,
respectively.


**Figures for appendix**

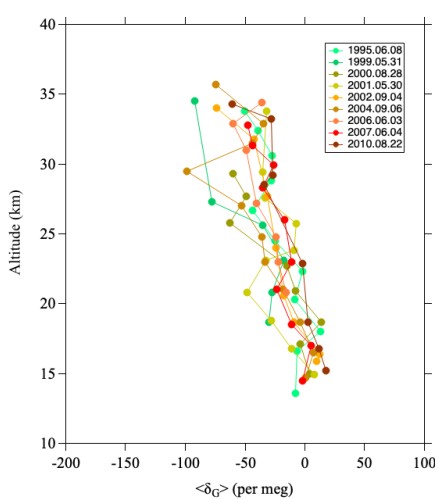

**Figure A1.** Vertical profiles of GS, $\langle\delta_G\rangle$, observed over Japan (Ishidoya et al., 2013).


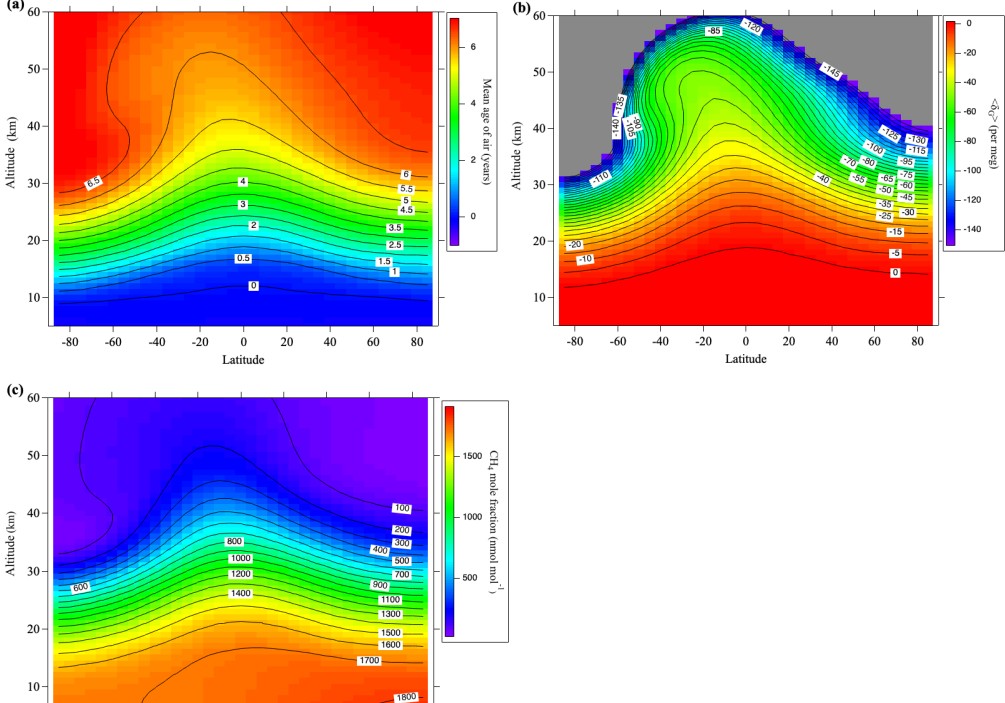

**Figure B1.** (**a**) Monthly average meridional distributions of the mean age of air for August 2016 calculated using the

SOCRATES model. (**b**) As in (**a**), but for gravitational separation, $\langle\delta_G\rangle$. Values lower than −150 per meg are shown in

gray. (**c**) As in (**a**), but for the $CH_4$ mole fraction.



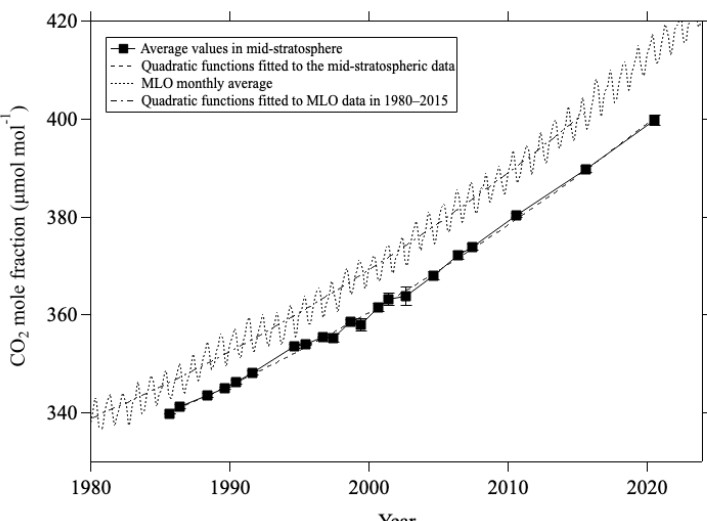

**Figure C1.** Average values of mid-stratospheric $CO_2$ mole fraction (>24 km altitude) over Japan (closed squares).
Monthly average $CO_2$ mole fractions at Mauna Loa observed by NOAA/ESRL are shown by the dotted line. Quadratic
functions fitted to the mid-stratospheric data (dashed line) and annual average MLO data from 1980 to 2015 (dashed-
dotted line) are also shown.

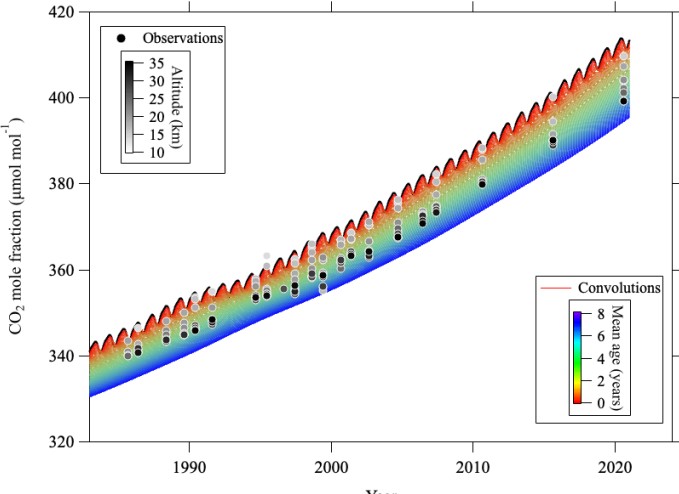

**Figure C2.** $CO_2$ mole fractions observed over Japan (circles) and convolutions (colored lines) calculated from the
tropospheric reference record and age spectrum. Observation altitudes of the $CO_2$ mole fraction are indicated by the gray
scale of the symbol colors. Note that the $CO_2$ mole fractions plotted have been corrected for $CH_4$ oxidation and GS.



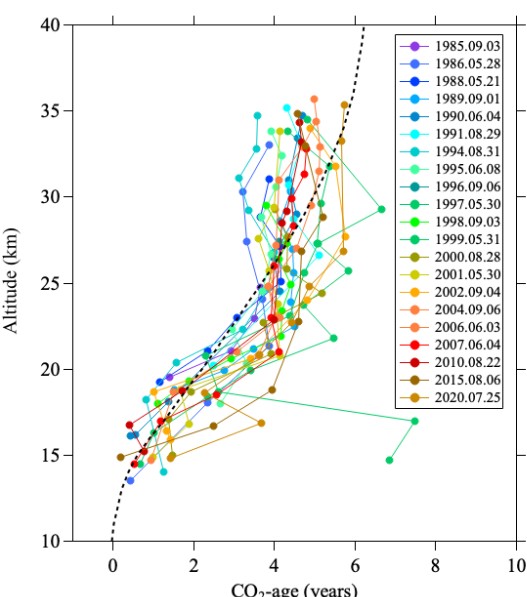

**Figure C3.** Vertical profiles of the $CO_2$ age calculated from $CO_2$ mole fractions from 1985 to 2020. The monthly average vertical profile of the mean age of air for August 2016 calculated using the SOCRATES model is shown by the dashed line.

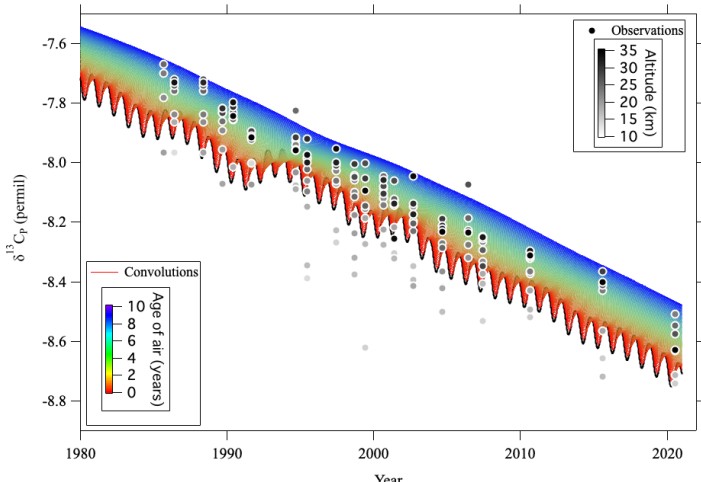

**Figure C4.** $\delta^{13}C_P$ observed over Japan (circles) and convolutions of $\delta^{13}C_P$ (colored lines) calculated from the tropospheric reference record and age spectrum. Observation altitudes of $\delta^{13}C_P$ are indicated by the gray scale of the symbol colors.