# Peer review of "Stratospheric $\delta^{13}CO_2$ observed over Japan and its governing processes"

_EGUsphere, 2025_

## Author Comment (AC1)

**Responses to the Referee #1 Comments**

Thank you very much for your significant and useful comments on the paper "Stratospheric $\delta^{13}CO_2$ observed over Japan and its governing processes and potential as an air age tracer" by Sugawara et al. We have revised the manuscript, considering your comments and suggestions. Details of our revision are as follows. The line numbers denote those of the revised manuscript.

*This manuscript presents a novel data set of stratospheric δ13CO2 dating back to 1985 and examines the processes controlling its distribution and its potential use as an age tracer. I think this is a very useful dataset. The results presented will be of interest to ACP readers, and I think they will be acceptable for publication after relatively minor changes.*

*My main concern with the manuscript is the discussion of the δ13CP age results. I think the authors are overstating the agreement with the CO2 age estimate and the potential use of the δ13CP age estimate. Specifically, in the abstract it is stated that "the mean age derived from δ13CP was consistent with that derived from the CO2 mole fraction, suggesting its usefulness for further investigation of stratospheric transport processes.", while in the conclusions it is started that "it [δ13CP] should prove a useful tool for investigating stratospheric transport processes" and "d13CP age is slightly larger than CO2 age ... by about 1.1 years on average".*

*The stated consistency occurs only because there is a very large (30%) uncertainty in δ13CP, and consistency is in terms of the mean over all data. Figure 10 shows that there is no statistical agreement for many individual measurements. Also, I don't think an average bias of 20% (which is what 1.1 yrs is) can be considered small. Given this large uncertainty and bias, I am doubtful that δ13CP age is a useful estimate/tool, as stated in the abstract and conclusions. It has the potential to do this only if the uncertainty can be greatly reduced.*

*I think the text needs to be modified to better indicate that there is large uncertainty and bias, and this needs to be reduced if this is to be a useful age tracer.*

We have deleted the descriptions about usefulness of $\delta^{13}C_P$ as an age tracer throughout the paper and clearly stated that the $\delta^{13}C_P$ age has large uncertainties and it cannot be used to improve mean age observations at present in Conclusions. Along with this, we have changed the manuscript title to "*Stratospheric $\delta^{13}CO_2$ observed over Japan and its governing processes*" by removing "*potential as an*

*air age tracer*". The related changes are as follows.

Lines 27-30: The sentences in Abstract "*used it to estimate the mean age of stratospheric air. Despite large uncertainties, the mean age derived from $\delta^{13}C_P$ was consistent with that derived from the $CO_2$ mole fraction, suggesting its usefulness for further investigation of stratospheric transport processes*" have been replaced with "*we found that $\delta^{13}C_P$ in the mid-latitude mid-stratosphere decreases over time with an about 5-year lag relative to the tropical upper troposphere. This fact strongly supports that stratospheric $\delta^{13}CO_2$ variations are governed by the airborne production of $^{13}C$-depleted $CO_2$ by $CH_4$ oxidation, the gravitational separation, and the propagation of the decreasing tropospheric $\delta^{13}CO_2$ trend into the stratosphere*"

Lines 505-524: We have removed many sentences about $\delta^{13}C_P$ age in Section 3.5 and moved the minimum necessary descriptions to Appendix C. We have deleted sentence "*However, the result that $\delta^{13}C_P$ age and $CO_2$ age are roughly consistent implies that $\delta^{13}C_P$ age could be used in addition to $SF_6$, halocarbons, etc., for better multi-component age estimations (Umezawa et al., 2024).*" and add "*At present, the $\delta^{13}C_P$ age is subject to larger uncertainties, making unsuitable for use as an additional constraint for age estimation. However, considering that the average $\delta^{13}C_P$ age was estimated to be 5.5 ± 1.6 years, the concept of stratospheric $\delta^{13}C_P$ itself would be valid.*".

Lines 539-541: We have deleted sentence in Conclusions "*our results showed that stratospheric $\delta^{13}C_P$ can serve as an additional age tracer alongside $CO_2$ and $SF_6$ mole fractions, and it should prove a useful tool for investigating stratospheric transport processes*" and add "*Because the $\delta^{13}C_P$ age has larger uncertainties than the $CO_2$ age at present, it is difficult to refine the mean age estimation. However, $\delta^{13}C_P$ in the mid-latitude mid-stratosphere decreased over time with a time delay and it was found to be quasi-conservative in the stratosphere.*".

*Following on from this, I think there could be more discussion of potential errors. On line 549 it is stated that "it is likely that d13CP age is overestimated because the effect of methane oxidation was underestimated in the calculation of d13CT." Why was methane oxidation underestimated, and can an estimate of the impact of this be made?*

Lines 681-687:We have added some descriptions about underestimation of the effect of methane oxidation in Appendix C. Note that the related sentences have been moved from Conclusions to Appendix C, in relation to the deletion of the descriptions about usefulness of $\delta^{13}C_P$ as an age tracer. We have added sentences "*The $\delta^{13}C_P$ age is larger than the $CO_2$ age (4.4 ± 0.6 years) by about 1.1 years on average. In*

this regard, $\delta^{13}C_T$ was calculated from the observed $CH_4$ mole fraction and its $\delta^{13}C$, assuming a closed system (Eq. 9). However, actual chemical processes do not occur in a closed system, and atmospheric mixing processes always result in apparent fractionation being smaller than true fractionation (Rahn et al. 1998; Kaiser et al. 2002; Toyoda et al., 2018). In this study, the isotopic effect of $CH_4$ oxidation was calculated based on the apparent fractionation factor. This would result in an underestimation of the $CH_4$ oxidation effect in Eq. 9 and an overestimation of the $\delta^{13}C_P$ age. To solve this problem, it is necessary to explicitly incorporate the isotope effect of $CH_4$ into the model, which will be a future challenge."
Along with this, we have added some references.

*Finally, the age calculation from δ13CP is buried at the end of Appendix C. Rather than having this appendix focused on CO2 and then meaning the new aspect (δ13CP) at the end, both tracers should be discussed at the start. Highlighting when differences in approach, or steps were could be large uncertainty.*

Lines 644-717: The title of Appendix C has been changed to "*Age calculation*" and totally rearranged descriptions following your suggestions. We first explained the age estimation methods common to both components and then described for $\delta^{13}C_P$ age. Figure of convolutions for $\delta^{13}C_P$ has been moved to C1. Accordingly, the order of the figures C1 – C4 has been changed.

Other revisions
Fig. 3
Because NOAA GML and INSTAAR have kindly provided us with updated $\delta^{13}C$ data at MLO in addition to well-organized data repository (Michel et al., 2025), we have replaced Fig.3 and replaced references. This change does not affect our results. Accordingly, the old organization name "ESRL" in NOAA was changed to the new name "GML".

---

## Author Comment (AC2)

**Responses to the Referee #2 Comments**

Thank you very much for your significant and useful comments on the paper "Stratospheric $\delta^{13}CO_2$ observed over Japan and its governing processes and potential as an air age tracer" by Sugawara et al. We have revised the manuscript, considering your comments and suggestions. Details of our revision are as follows. The line numbers denote those of the revised manuscript.

*This paper uses flask sample measurements from balloon flights in the stratosphere over Japan to investigate the stable carbon isotopic ratio of CO2 and other molecules involved in CO2 photochemistry in the stratosphere. The measurements and analytical methods are described in some detail and as far as I could follow it the techniques seemed reasonable, but I am not an expert in this topic as the authors clearly are. The discussions of the mechanisms responsible for the delta13CO2 and CO2 stratospheric profiles are well done and the schematic shown in Figure 6 provides a nice summary of the competing processes. The 2D modeling is interesting in general and provides some support for the mechanisms described here as the primary drivers of the observed distributions. The model transport is inaccurate, as the authors attest, but that is a common limitation of most stratospheric models.*

*The use of delta13Cp as an age of air tracer is somewhat dubious. The large uncertainties on the ages with this quantity make it essentially unusable on its own. The authors do suggest that it could be used in combination with other trace gas measurements for multi-component age estimates but it seems unlikely that this new quantity will help constrain any of the current age of air trace gas estimates. My main suggestion would be to considerably shorten Section 3.5 by removing much of the detail of the age of air calculation with delta13Cp.*

We have deleted the descriptions about usefulness of $\delta^{13}C_P$ as an age tracer throughout the paper and clearly stated that the $\delta^{13}C_P$ age has large uncertainties and it cannot be used to improve mean age observations at present in Conclusions. Along with this, we have changed the manuscript title to "*Stratospheric $\delta^{13}CO_2$ observed over Japan and its governing processes*" by removing "*potential as an air age tracer*". The related changes are as follows.

Lines 27-30: The sentences in Abstract "*used it to estimate the mean age of stratospheric air. Despite large uncertainties, the mean age derived from $\delta^{13}C_P$ was consistent with that derived from the $CO_2$ mole fraction, suggesting its usefulness for further investigation of stratospheric transport processes*" have been replaced with "*we found that $\delta^{13}C_P$ in the mid-latitude mid-stratosphere decreases over time*

with an about 5-year lag relative to the tropical upper troposphere. This fact strongly supports that stratospheric $\delta^{13}CO_2$ variations are governed by the airborne production of $^{13}C$-depleted $CO_2$ by $CH_4$ oxidation, the gravitational separation, and the propagation of the decreasing tropospheric $\delta^{13}CO_2$ trend into the stratosphere"

Lines 505-524: We have removed many sentences about $\delta^{13}C_P$ age in Section 3.5 and moved the minimum necessary descriptions to Appendix C. We have deleted sentence "*However, the result that $\delta^{13}C_P$ age and $CO_2$ age are roughly consistent implies that $\delta^{13}C_P$ age could be used in addition to $SF_6$, halocarbons, etc., for better multi-component age estimations (Umezawa et al., 2024).*" and add "*At present, the $\delta^{13}C_P$ age is subject to larger uncertainties, making unsuitable for use as an additional constraint for age estimation. However, considering that the average $\delta^{13}C_P$ age was estimated to be $5.5 \pm 1.6$ years, the concept of stratospheric $\delta^{13}C_P$ itself would be valid.*".

Lines 539-541: We have deleted sentence in Conclusions "*our results showed that stratospheric $\delta^{13}C_P$ can serve as an additional age tracer alongside $CO_2$ and $SF_6$ mole fractions, and it should prove a useful tool for investigating stratospheric transport processes*" and add "*Because the $\delta^{13}C_P$ age has larger uncertainties than the $CO_2$ age at present, it is difficult to refine the mean age estimation. However, $\delta^{13}C_P$ in the mid-latitude mid-stratosphere decreased over time with a time delay and it was found to be quasi-conservative in the stratosphere.*".

Lines 644-717: The title of Appendix C has been changed to "*Age calculation*" and totally rearranged descriptions following suggestions by RC1. Details of $\delta^{13}C_P$ age calculation have been partly moved from Sect. 3.5 to Appendix C. Accordingly, the order of the figures C1 – C4 has been changed.

*Overall, this paper presents novel measurements and analysis suitable for publication in ACP. I recommend publication with consideration of the main comment above and the specific comments below.*

*Specific comments:*
*Line 18: add 'the' after 'investigate'*

As described above, we revised the abstract and removed a sentence "*and to investigate the usefulness of $\delta^{13}CO_2$ as an age tracer,* ".

*Line 45: For those of us not as familiar with the details of isotopic studies, perhaps a brief explanation of MIE would be helpful here.*

Lines 47-50: We added sentences as follows:

"*The isotopic fractionation is usually caused by the effect of mass differences between isotopologues (i.e. the mass-dependent isotopic effect). In this case, the fractionation between $^{18}O$ and $^{16}O$ is almost twice as large as that between $^{17}O$ and $^{16}O$. However, the isotopic fractionation that does not follow this relation occurs mainly in photochemical processes.*"

*Figure 2c: Appear to be missing the dotted line in this figure that would show the model results without the tropospheric trend, GS or airborne sources.*

Figure 2c: The dotted line has been changed to a thicker dotted line to make it easier to see. Because that line shows a result without the tropospheric trend, GS or airborne sources, the vertical distribution becomes constant. Figure caption has been revised as follows:

"*Curves show 2-D modeling results without the tropospheric trend, seasonal cycle, GS, or airborne sources (thick dotted line), …*"

*Lines 380-5: The GS correction of CO2 is interesting, although not as significant as for delta13CO2 as is mentioned. The example of 22 August 2010 is said to have a GS correction of 0.4 ppm for CO2 at the 34 km altitude level but it doesn't look that large in Fig. 5a. Certainly, all of the points below the 34 km level have very small GS corrections for CO2. But 0.4-0.6 ppm would have an effect on the age of air calculation with CO2. Would you recommend that all age of air calculations with CO2 use a GS correction? If so, would an average profile of deltaG as in Fig. A1 be appropriate to use in Eqn. 8? The implication here is that age of air from CO2 without taking into account GS, which is essentially all age of air calculations done thus far, has an old age bias that increases with height. If this bias is quite small, say less than a month, then this is not significant. But it appears to be larger than that at high altitudes. A brief statement here about age of air implications would be useful.*

As you pointed out, we recommend that GS corrections will be applied not only for $\delta^{13}C$ but also for $CO_2$ mole fraction in order to estimate the mean ages more precisely. Our $CO_2$ age was calculated with taking into account GS, as described in Appendix C.

Lines 390-393: As you suggested, we added some sentences as follows:

"*In this regard, the mean age estimated from the CO2 mole fraction without GS correction has an older age*

*bias that increases with increasing altitude. The age bias is negligibly small (< 1 month) in the lower stratosphere but not negligible in the mid-stratosphere ( > 2 months at 35 km altitude). Therefore, we applied the GS correction to the $CO_2$ mole fraction to estimate the $CO_2$ age, as described in Appendix C."*

*Figure 10: I would suggest removing the trend line for the delta13Cp age since the uncertainty on the values are large and the trend is not significant. The insignificance of the trend is mentioned in the text but the figure implies there is a discrepancy in the age trends rather than that they are in agreement within uncertainties.*

Figure 10: We removed the trend line for $\delta^{13}C_P$ age, as you suggested. Figure caption has been corrected to "*Dashed line is a linear least-squares fit to the $CO_2$ ages. The linear fit to the $\delta^{13}C_P$ age is not shown because the trend is not significant.*"

*Line 598: add 'is' before 'important'*

Line 584: We corrected it. Thank you.

Other revisions
Fig. 3
Because NOAA GML and INSTAAR have kindly provided us with updated $\delta^{13}C$ data at MLO in addition to well-organized data repository (Michel et al., 2025), we have replaced Fig.3 and replaced references. This change does not affect our results. Accordingly, the old organization name "ESRL" in NOAA was changed to the new name "GML".

---

## Author Response (AR2)

**Author Response to the Referee #2**

Thank you very much for your significant and useful comments on the paper "Stratospheric $\delta^{13}CO_2$ observed over Japan and its governing processes" by Sugawara et al. We have revised the manuscript, considering your comments and suggestions. Details of our revision are as follows. The line numbers denote those of the revised manuscript.

*Lines 547-8: I would suggest clarifying this sentence to something like: '..delta13Cp age has a relatively large uncertainty, making it unsuitable for use as an additional constraint for the age estimation.'*

Lines 522-523: We have replaced sentence "*the $\delta^{13}C_P$ age is subject to larger uncertainties, making unsuitable for use as an additional constraint for age estimation*" with "*the $\delta^{13}C_P$ age has a relatively large uncertainty, making it unsuitable for use as an additional constraint for the age estimation*".